# Accelerating Convergence of Replica Exchange Stochastic Gradient MCMC via Variance Reduction

**Wei Deng** *
Department of Mathematics
Purdue University
West Lafayette, IN, USA
weideng056@gmail.com

**Qi Feng** *
Department of Mathematics
University of Southern California
Los Angeles, CA, USA
qif@usc.edu

**Georgios Karagiannis**
Department of Mathematical Sciences
Durham University
Durham, UK
georgios.karagiannis@durham.ac.uk

**Guang Lin**
Departments of Mathematics &
School of Mechanical Engineering
Purdue University
West Lafayette, IN, USA
guanglin@purdue.edu

**Faming Liang**
Departments of Statistics
Purdue University
West Lafayette, IN, USA
fmliang@purdue.edu

## Abstract

Replica exchange stochastic gradient Langevin dynamics (reSGLD) has shown promise in accelerating the convergence in non-convex learning; however, an excessively large correction for avoiding biases from noisy energy estimators has limited the potential of the acceleration. To address this issue, we study the variance reduction for noisy energy estimators, which promotes much more effective swaps. Theoretically, we provide a non-asymptotic analysis on the exponential convergence for the underlying continuous-time Markov jump process; moreover, we consider a generalized Girsanov theorem which includes the change of Poisson measure to overcome the crude discretization based on the Grönwall's inequality and yields a much tighter error in the 2-Wasserstein ($\mathcal{W}_2$) distance. Numerically, we conduct extensive experiments and obtain state-of-the-art results in optimization and uncertainty estimates for synthetic experiments and image data.

## 1 Introduction

Stochastic gradient Monte Carlo methods (Welling & Teh, 2011; Chen et al., 2014; Li et al., 2016) are the golden standard for Bayesian inference in deep learning due to their theoretical guarantees in uncertainty quantification (Vollmer et al., 2016; Chen et al., 2015) and non-convex optimization (Zhang et al., 2017). However, despite their scalability with respect to the data size, their mixing rates are often extremely slow for complex deep neural networks with rugged energy landscapes (Li et al., 2018). To speed up the convergence, several techniques have been proposed in the literature in order to accelerate their exploration of multiple modes on the energy landscape, for example, dynamic temperatures (Ye et al., 2017) and cyclic learning rates (Zhang et al., 2020), to name a few. However, such strategies only explore contiguously a limited region around a few informative modes. Inspired by the successes of replica exchange, also known as parallel tempering, in traditional Monte Carlo methods (Swendsen & Wang, 1986; Earl & Deem, 2005), reSGLD (Deng et al.,

---

*Equal contribution

2020) uses multiple processes based on stochastic gradient Langevin dynamics (SGLD) where interactions between different SGLD chains are conducted in a manner that encourages large jumps. In addition to the ideal utilization of parallel computation, the resulting process is able to jump to more informative modes for more robust uncertainty quantification. However, the noisy energy estimators in mini-batch settings lead to a large bias in the naïve swaps, and a large correction is required to reduce the bias, which yields few effective swaps and insignificant accelerations. Therefore, how to reduce the variance of noisy energy estimators becomes essential in speeding up the convergence.

A long standing technique for variance reduction is the control variates method. The key to reducing the variance is to properly design correlated control variates so as to counteract some noise. Towards this direction, Dubey et al. (2016); Xu et al. (2018) proposed to update the control variate periodically for the stochastic gradient estimators and Baker et al. (2019) studied the construction of control variates using local modes. Despite the advantages in near-convex problems, a natural discrepancy between theory (Chatterji et al., 2018; Xu et al., 2018; Zou et al., 2019b) and practice (He et al., 2016; Devlin et al., 2019) is *whether we should avoid the gradient noise in non-convex problems*. To fill in the gap, we only focus on the variance reduction of noisy energy estimators to exploit the theoretical accelerations but no longer consider the variance reduction of the noisy gradients so that the empirical experience from stochastic gradient descents with momentum (M-SGD) can be naturally imported.

In this paper we propose the variance-reduced replica exchange stochastic gradient Langevin dynamics (VR-reSGLD) algorithm to accelerate convergence by reducing the variance of the noisy energy estimators. This algorithm not only *shows the potential of exponential acceleration* via much more effective swaps in the non-asymptotic analysis but also *demonstrates remarkable performance in practical tasks* where a limited time is required; while others (Xu et al., 2018; Zou et al., 2019a) may only work well when the dynamics is sufficiently mixed and the discretization error becomes a major component. Moreover, the existing discretization error of the Langevin-based Markov jump processes (Chen et al., 2019; Deng et al., 2020; Futami et al., 2020) is exponentially dependent on time due to the limitation of Grönwall's inequality. To avoid such a crude estimate, we consider the generalized Girsanov theorem and a change of Poisson measure. As a result, we obtain a much *tighter discretization error only polynomially dependent on time*. Empirically, we test the algorithm through extensive experiments and achieve state-of-the-art performance in both optimization and uncertainty estimates.

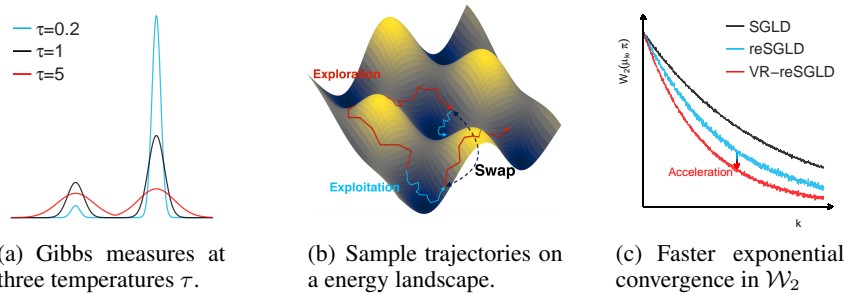

(a) Gibbs measures at three temperatures $\tau$.

(b) Sample trajectories on a energy landscape.

(c) Faster exponential convergence in $\mathcal{W}_2$

Figure 1: An illustration of replica exchange Monte Carlo algorithms for non-convex learning.

## 2 PRELIMINARIES

A common problem, in Bayesian inference, is the simulation from a posterior $P(\boldsymbol{\beta}|\boldsymbol{X}) \propto P(\boldsymbol{\beta}) \prod_{i=1}^{N} P(\mathbf{x}_i|\boldsymbol{\beta})$, where $P(\boldsymbol{\beta})$ is a proper prior, $\prod_{i=1}^{N} P(\mathbf{x}_i|\boldsymbol{\beta})$ is the likelihood function and $N$ is the number of data points. When $N$ is large, the standard Langevin dynamics is too costly in evaluating the gradients. To tackle this issue, stochastic gradient Langevin dynamics (SGLD) (Welling & Teh, 2011) was proposed to make the algorithm scalable by approximating the gradient through a mini-batch data $B$ of size $n$ such that

$$\boldsymbol{\beta}_k = \boldsymbol{\beta}_{k-1} - \eta_k \frac{N}{n} \sum_{i \in B_k} \nabla L(\mathbf{x}_i|\boldsymbol{\beta}_{k-1}) + \sqrt{2\eta_k \tau} \boldsymbol{\xi}_k, \tag{1}$$

where $\boldsymbol{\beta}_k \in \mathbb{R}^d$, $\tau$ denotes the temperature, $\eta_k$ is the learning rate at iteration $k$, $\boldsymbol{\xi}_k$ is a standard Gaussian vector, and $L(\cdot) := -\log \mathrm{P}(\boldsymbol{\beta}|\boldsymbol{X})$ is the energy function. SGLD is known to converge weakly to a stationary Gibbs measure $\pi_\tau(\boldsymbol{\beta}) \propto \exp\left(-L(\boldsymbol{\beta})/\tau\right)$ as $\eta_k$ decays to 0 (Teh et al., 2016).

The temperature $\tau$ is the key to accelerating the computations in multi-modal distributions. On the one hand, a high temperature flattens the Gibbs distribution $\exp\left(-L(\boldsymbol{\beta})/\tau\right)$ (see the red curve in Fig.1(a)) and accelerates mixing by facilitating exploration of the whole domain, but the resulting distribution becomes much less concentrated around the global optima. On the other hand, a low temperature exploits the local region rapidly; however, it may cause the particles to stick in a local region for an exponentially long time, as shown in the blue curve in Fig.1(a,b). To bridge the gap between global exploration and local exploitation, Deng et al. (2020) proposed the replica exchange SGLD algorithm (reSGLD), which consists of a low-temperature SGLD to encourage exploitation and a high-temperature SGLD to support exploration

$$\boldsymbol{\beta}_k^{(1)} = \boldsymbol{\beta}_{k-1}^{(1)} - \eta_k \frac{N}{n} \sum_{i \in B_k} \nabla L(\mathbf{x}_i|\boldsymbol{\beta}_{k-1}^{(1)}) + \sqrt{2\eta_k \tau^{(1)}}\boldsymbol{\xi}_k^{(1)}$$

$$\boldsymbol{\beta}_k^{(2)} = \boldsymbol{\beta}_{k-1}^{(2)} - \eta_k \frac{N}{n} \sum_{i \in B_k} \nabla L(\mathbf{x}_i|\boldsymbol{\beta}_{k-1}^{(2)}) + \sqrt{2\eta_k \tau^{(2)}}\boldsymbol{\xi}_k^{(2)},$$

where the invariant measure is known to be $\pi(\boldsymbol{\beta}^{(1)}, \boldsymbol{\beta}^{(2)}) \propto \exp\left(-\frac{L(\boldsymbol{\beta}^{(1)})}{\tau^{(1)}} - \frac{L(\boldsymbol{\beta}^{(2)})}{\tau^{(2)}}\right)$ as $\eta_k \to 0$ and $\tau^{(1)} < \tau^{(2)}$. Moreover, the two processes may swap the positions to allow tunneling between different modes. To avoid inducing a large bias in mini-batch settings, a corrected swapping rate $\widehat{S}$ is developed such that

$$\widehat{S} = \exp\left\{ \left(\frac{1}{\tau^{(1)}} - \frac{1}{\tau^{(2)}}\right) \left(\frac{N}{n} \sum_{i \in B_k} L(\mathbf{x}_i|\boldsymbol{\beta}_k^{(1)}) - \frac{N}{n} \sum_{i \in B_k} L(\mathbf{x}_i|\boldsymbol{\beta}_k^{(2)}) - \frac{\left(\frac{1}{\tau^{(1)}} - \frac{1}{\tau^{(2)}}\right)\widehat{\sigma}^2}{F}\right)\right\},$$

where $\widehat{\sigma}^2$ is an estimator of the variance of $\frac{N}{n}\sum_{i \in B_k} L(\mathbf{x}_i|\boldsymbol{\beta}_k^{(1)}) - \frac{N}{n}\sum_{i \in B_k} L(\mathbf{x}_i|\boldsymbol{\beta}_k^{(2)})$ and $F$ is the correction factor to balance between acceleration and bias. In other words, the parameters switch the positions from $(\boldsymbol{\beta}_k^{(1)}, \boldsymbol{\beta}_k^{(2)})$ to $(\boldsymbol{\beta}_k^{(2)}, \boldsymbol{\beta}_k^{(1)})$ with a probability $r(1 \wedge \widehat{S})\eta_k$, where the constant $r$ is the swapping intensity and can set to $\frac{1}{\eta_k}$ for simplicity.

From a probabilistic point of view, reSGLD is a discretization scheme of replica exchange Langevin diffusion (reLD) in mini-batch settings. Given a smooth test function $f$ and a swapping-rate function $S$, the infinitesimal generator $\mathcal{L}_S$ associated with the continuous-time reLD follows

$$\mathcal{L}_S f(\boldsymbol{\beta}^{(1)}, \boldsymbol{\beta}^{(2)}) = -\langle \nabla_{\boldsymbol{\beta}^{(1)}} f(\boldsymbol{\beta}^{(1)}, \boldsymbol{\beta}^{(2)}), \nabla L(\boldsymbol{\beta}^{(1)})\rangle - \langle \nabla_{\boldsymbol{\beta}^{(2)}} f(\boldsymbol{\beta}^{(1)}, \boldsymbol{\beta}^{(2)}), \nabla L(\boldsymbol{\beta}^{(2)})\rangle$$
$$+ \tau^{(1)}\Delta_{\boldsymbol{\beta}^{(1)}} f(\boldsymbol{\beta}^{(1)}, \boldsymbol{\beta}^{(2)}) + \tau^{(2)}\Delta_{\boldsymbol{\beta}^{(2)}} f(\boldsymbol{\beta}^{(1)}, \boldsymbol{\beta}^{(2)}) + rS(\boldsymbol{\beta}^{(1)}, \boldsymbol{\beta}^{(2)}) \cdot (f(\boldsymbol{\beta}^{(2)}, \boldsymbol{\beta}^{(1)}) - f(\boldsymbol{\beta}^{(1)}, \boldsymbol{\beta}^{(2)})),$$

where the last term arises from swaps and $\Delta_{\boldsymbol{\beta}^{(\cdot)}}$ is the the Laplace operator with respect to $\boldsymbol{\beta}^{(\cdot)}$. Note that the infinitesimal generator is closely related to Dirichlet forms in characterizing the evolution of a stochastic process. By standard calculations in Markov semigroups (Chen et al., 2019), the Dirichlet form $\mathcal{E}_S$ associated with the infinitesimal generator $\mathcal{L}_S$ follows

$$\mathcal{E}_S(f) = \underbrace{\int \left(\tau^{(1)}\|\nabla_{\boldsymbol{\beta}^{(1)}} f(\boldsymbol{\beta}^{(1)}, \boldsymbol{\beta}^{(2)})\|^2 + \tau^{(2)}\|\nabla_{\boldsymbol{\beta}^{(2)}} f(\boldsymbol{\beta}^{(1)}, \boldsymbol{\beta}^{(2)})\|^2\right) d\pi(\boldsymbol{\beta}^{(1)}, \boldsymbol{\beta}^{(2)})}_{\text{vanilla term } \mathcal{E}(f)}$$
$$+ \underbrace{\frac{r}{2}\int S(\boldsymbol{\beta}^{(1)}, \boldsymbol{\beta}^{(2)}) \cdot (f(\boldsymbol{\beta}^{(2)}, \boldsymbol{\beta}^{(1)}) - f(\boldsymbol{\beta}^{(1)}, \boldsymbol{\beta}^{(2)}))^2 d\pi(\boldsymbol{\beta}^{(1)}, \boldsymbol{\beta}^{(2)})}_{\text{acceleration term}}, \tag{2}$$

which leads to a strictly positive acceleration under mild conditions and is crucial for the exponentially accelerated convergence in the $\mathcal{W}_2$ distance (see Fig.1(c)). However, the acceleration depends on the swapping-rate function $S$ and becomes much smaller given a noisy estimate of $\frac{N}{n}\sum_{i \in B} L(\mathbf{x}_i|\boldsymbol{\beta})$ due to the demand of large corrections to reduce the bias.

## 3 VARIANCE REDUCTION IN REPLICA EXCHANGE STOCHASTIC GRADIENT LANGEVIN DYNAMICS

The desire to obtain more effective swaps and larger accelerations drives us to design more efficient energy estimators. A naïve idea would be to apply a large batch size $n$, which reduces the variance of the noisy energy estimator proportionally. However, this comes with a significantly increased memory overhead and computations and therefore is inappropriate for big data problems.

A natural idea to propose more effective swaps is to reduce the variance of the noisy energy estimator $L(B|\boldsymbol{\beta}^{(h)}) = \frac{N}{n}\sum_{i\in B} L(\mathbf{x}_i|\boldsymbol{\beta}^{(h)})$ for $h \in \{1, 2\}$. Considering an unbiased estimator $L(B|\widehat{\boldsymbol{\beta}}^{(h)})$ for $\sum_{i=1}^N L(\mathbf{x}_i|\widehat{\boldsymbol{\beta}}^{(h)})$ and a constant $c$, we see that a new estimator $\widetilde{L}(B|\boldsymbol{\beta}^{(h)})$, which follows

$$\widetilde{L}(B|\boldsymbol{\beta}^{(h)}) = L(B|\boldsymbol{\beta}^{(h)}) + c\left(L(B|\widehat{\boldsymbol{\beta}}^{(h)}) - \sum_{i=1}^N L(\mathbf{x}_i|\widehat{\boldsymbol{\beta}}^{(h)})\right), \tag{3}$$

is still the unbiased estimator for $\sum_{i=1}^N L(\mathbf{x}_i|\boldsymbol{\beta}^{(h)})$. By decomposing the variance, we have

$$\mathrm{Var}(\widetilde{L}(B|\boldsymbol{\beta}^{(h)})) = \mathrm{Var}\left(L(B|\boldsymbol{\beta}^{(h)})\right) + c^2\mathrm{Var}\left(L(B|\widehat{\boldsymbol{\beta}}^{(h)})\right) + 2c\mathrm{Cov}\left(L(B|\boldsymbol{\beta}^{(h)}), L(B|\widehat{\boldsymbol{\beta}}^{(h)})\right).$$

In such a case, $\mathrm{Var}(\widetilde{L}(B|\boldsymbol{\beta}^{(h)}))$ achieves the minimum variance $(1 - \rho^2)\mathrm{Var}(L(B|\boldsymbol{\beta}^{(h)}))$ given $c^\star := -\frac{\mathrm{Cov}(L(B|\boldsymbol{\beta}^{(h)}), L(B|\widehat{\boldsymbol{\beta}}^{(h)}))}{\mathrm{Var}(L(B|\widehat{\boldsymbol{\beta}}^{(h)}))}$, where $\mathrm{Cov}(\cdot, \cdot)$ denotes the covariance and $\rho$ is the correlation coefficient of $L(B|\boldsymbol{\beta}^{(h)})$ and $L(B|\widehat{\boldsymbol{\beta}}^{(h)})$. To propose a correlated control variate, we follow Johnson & Zhang (2013) and update $\widehat{\boldsymbol{\beta}}^{(h)} = \boldsymbol{\beta}^{(h)}_{m\lfloor\frac{k}{m}\rfloor}$ every $m$ iterations. Moreover, the optimal $c^\star$ is often unknown in practice. To handle this issue, a well-known solution (Johnson & Zhang, 2013) is to fix $c = -1$ given a high correlation $|\rho|$ of the estimators and then we can present the VR-reSGLD algorithm in Algorithm 1. Since the exact variance for correcting the stochastic swapping rate is unknown and even time-varying, we follow Deng et al. (2020) and propose to use stochastic approximation (Robbins & Monro, 1951) to adaptively update the unknown variance.

**Variants of VR-reSGLD**  The number of iterations $m$ to update the control variate $\widehat{\boldsymbol{\beta}}^{(h)}$ gives rise to a trade-off in computations and variance reduction. A small $m$ introduces a highly correlated control variate at the cost of expensive computations; a large $m$, however, may yield a less correlated control variate and setting $c = -1$ fails to reduce the variance. In spirit of the adaptive variance in Deng et al. (2020) to estimate the unknown variance, we explore the idea of the adaptive coefficient $\widetilde{c}_k = (1 - \gamma_k)\widetilde{c}_{k-m} + \gamma_k c_k$ such that the unknown optimal $c^\star$ is well approximated. We present the adaptive VR-reSGLD in Algorithm 2 in Appendix E.2 and show empirically later that the adaptive VR-reSGLD leads to a significant improvement over VR-reSGLD for the less correlated estimators.

A parallel line of research is to exploit the SAGA algorithm (Defazio et al., 2014) in the study of variance reduction. Despite the most effective performance in variance reduction (Chatterji et al., 2018), the SAGA type of sampling algorithms require an excessively memory storage of $\mathcal{O}(Nd)$, which is too costly for big data problems. Therefore, we leave the study of the lightweight SAGA algorithm inspired by Harikandeh et al. (2015); Zhou et al. (2019) for future works.

**Related work**  Although our VR-reSGLD is, in spirit, similar to VR-SGLD (Dubey et al., 2016; Xu et al., 2018), it differs from VR-SGLD in two aspects: First, VR-SGLD conducts variance reduction on the gradient and only shows promises in the nearly log-concave distributions or when the Markov process is sufficiently converged; however, our VR-reSGLD solely focuses on the variance reduction of the energy estimator to propose more effective swaps, and therefore we can import the empirical experience in hyper-parameter tuning from M-SGD to our proposed algorithm. Second, VR-SGLD doesn't accelerate the continuous-time Markov process but only focuses on reducing the discretization error; VR-reSGLD possesses a larger acceleration term in the Dirichlet form (2) and shows a potential in exponentially speeding up the convergence of the continuous-time process in the early stage, in addition to the improvement on the discretization error. In other words, our algorithm is not only theoretically sound but also more empirically appealing for a wide variety of problems in non-convex learning.

---

**Algorithm 1** Variance-reduced replica exchange stochastic gradient Langevin dynamics (VR-reSGLD). The learning rate and temperature can be set to dynamic to speed up the computations. A larger smoothing factor $\gamma$ captures the trend better but becomes less robust. $\mathbb{T}$ is the thinning factor to avoid a cumbersome system.

---

**Input** The initial parameters $\boldsymbol{\beta}_0^{(1)}$ and $\boldsymbol{\beta}_0^{(2)}$, learning rate $\eta$, temperatures $\tau^{(1)}$ and $\tau^{(2)}$, correction factor $F$ and smoothing factor $\gamma$.
**repeat**
    **Parallel sampling** Randomly pick a mini-batch set $B_k$ of size $n$.

$$\boldsymbol{\beta}_k^{(h)} = \boldsymbol{\beta}_{k-1}^{(h)} - \eta \frac{N}{n} \sum_{i \in B_k} \nabla L(\mathbf{x}_i | \boldsymbol{\beta}_{k-1}^{(h)}) + \sqrt{2\eta \tau^{(h)}} \boldsymbol{\xi}_k^{(h)}, \text{ for } h \in \{1, 2\}. \tag{4}$$

    **Variance-reduced energy estimators** Update $\widehat{L}^{(h)} = \sum_{i=1}^N L\left(\mathbf{x}_i | \boldsymbol{\beta}_{m\lfloor \frac{k}{m} \rfloor}^{(h)}\right)$ every $m$ iterations.

$$\widetilde{L}(B_k | \boldsymbol{\beta}_k^{(h)}) = \frac{N}{n} \sum_{i \in B_k} \left[ L(\mathbf{x}_i | \boldsymbol{\beta}_k^{(h)}) - L\left(\mathbf{x}_i \Big| \boldsymbol{\beta}_{m\lfloor \frac{k}{m} \rfloor}^{(h)}\right) \right] + \widehat{L}^{(h)}, \text{ for } h \in \{1, 2\}. \tag{5}$$

    **if** $k \bmod m = 0$ **then**
        Update $\widetilde{\sigma}_k^2 = (1 - \gamma)\widetilde{\sigma}_{k-m}^2 + \gamma \sigma_k^2$, where $\sigma_k^2$ is an estimate for $\text{Var}\left(\widetilde{L}(B_k | \boldsymbol{\beta}_k^{(1)}) - \widetilde{L}(B_k | \boldsymbol{\beta}_k^{(2)})\right)$.
    **end if**
    **Bias-reduced swaps** Swap $\boldsymbol{\beta}_{k+1}^{(1)}$ and $\boldsymbol{\beta}_{k+1}^{(2)}$ if $u < \widetilde{S}_{\eta,m,n}$, where $u \sim \text{Unif}\,[0,1]$, and $\widetilde{S}_{\eta,m,n}$ follows

$$\widetilde{S}_{\eta,m,n} = \exp\left\{ \left(\frac{1}{\tau^{(1)}} - \frac{1}{\tau^{(2)}}\right) \left(\widetilde{L}(B_{k+1} | \boldsymbol{\beta}_{k+1}^{(1)}) - \widetilde{L}(B_{k+1} | \boldsymbol{\beta}_{k+1}^{(2)}) - \frac{1}{F}\left(\frac{1}{\tau^{(1)}} - \frac{1}{\tau^{(2)}}\right) \widetilde{\sigma}_{m\lfloor \frac{k}{m} \rfloor}^2\right) \right\}. \tag{6}$$

**until** $k = k_{\max}$.
**Output:** The low-temperature process $\{\boldsymbol{\beta}_{i\mathbb{T}}^{(1)}\}_{i=1}^{\lfloor k_{\max}/\mathbb{T} \rfloor}$, where $\mathbb{T}$ is the thinning factor.

---

## 4 THEORETICAL PROPERTIES

The large variance of noisy energy estimators directly limits the potential of the acceleration and significantly slows down the convergence compared to the replica exchange Langevin dynamics. As a result, VR-reSGLD may lead to a more efficient energy estimator with a much smaller variance.

**Lemma 1 (Variance-reduced energy estimator)** *Under the smoothness and dissipativity assumptions 1 and 2 in Appendix A, the variance of the variance-reduced energy estimator $\widetilde{L}(B | \boldsymbol{\beta}^{(h)})$, where $h \in \{1, 2\}$, is upper bounded by*

$$\text{Var}\left(\widetilde{L}(B | \boldsymbol{\beta}^{(h)})\right) \leq \min \left\{ \mathcal{O}\left(\frac{m^2 \eta}{n}\right), \text{Var}\left(\frac{N}{n} \sum_{i \in B} L(\mathbf{x}_i | \boldsymbol{\beta}^{(h)})\right) + \text{Var}\left(\frac{N}{n} \sum_{i \in B} L(\mathbf{x}_i | \widehat{\boldsymbol{\beta}}^{(h)})\right) \right\},$$

*where the detailed $\mathcal{O}(\cdot)$ constants is shown in Lemma B1 in the appendix.*

The analysis shows the variance-reduced estimator $\widetilde{L}(B | \boldsymbol{\beta}^{(h)})$ yields a much-reduced variance given a smaller learning rate $\eta$ and a smaller $m$ for updating control variates based on the batch size $n$. Although the truncated swapping rate $S_{\eta,m,n} = \min\{1, \widetilde{S}_{\eta,m,n}\}$ still satisfies the "stochastic" detailed balance given an unbiased swapping-rate estimator $\widetilde{S}_{\eta,m,n}$ (Deng et al., 2020) [†], it doesn't mean the efficiency of the swaps is not affected. By contrast, we can show that the number of swaps may become *exponentially smaller on average*.

**Lemma 2 (Variance reduction for larger swapping rates)** *Given a large enough batch size $n$, the variance-reduced energy estimator $\widetilde{L}(B_k | \boldsymbol{\beta}_k^{(h)})$ yields a truncated swapping rate that satisfies*

$$\mathbb{E}[S_{\eta,m,n}] \approx \min\left\{ 1, S(\boldsymbol{\beta}^{(1)}, \boldsymbol{\beta}^{(2)})\left(\mathcal{O}\left(\frac{1}{n^2}\right) + e^{-\mathcal{O}\left(\frac{m^2 \eta}{n} + \frac{1}{n^2}\right)}\right) \right\}, \tag{7}$$

---

[†] Andrieu & Roberts (2009); Quiroz et al. (2019) achieve a similar result based on the unbiased likelihood estimator for the Metropolis-hasting algorithm. See section 3.1 (Quiroz et al., 2019) for details.

where $S(\boldsymbol{\beta}^{(1)}, \boldsymbol{\beta}^{(2)})$ is the deterministic swapping rate defined in Appendix B. The proof is shown in Lemma.B2 in Appendix B. Note that the above lemma doesn't require the normality assumption. As $n$ goes to infinity, where the asymptotic normality holds, the RHS of (7) changes to $\min\left\{1, S(\boldsymbol{\beta}^{(1)}, \boldsymbol{\beta}^{(2)})e^{-\mathcal{O}\left(\frac{m^2\eta}{n}\right)}\right\}$, which becomes exponentially larger as we use a smaller update frequency $m$ and learning rate $\eta$. Since the continuous-time reLD induces a jump operator in the infinitesimal generator, the resulting Dirichlet form potentially leads to a much larger acceleration term which linearly depends on the swapping rate $S_{\eta,m,n}$ and yields a faster exponential convergence. Now we are ready to present the first main result.

**Theorem 1 (Exponential convergence)** *Under the smoothness and dissipativity assumptions 1 and 2, the probability measure associated with reLD at time $t$, denoted as $\nu_t$, converges exponentially fast to the invariant measure $\pi$:*

$$\mathcal{W}_2(\nu_t, \pi) \leq D_0 \exp\left\{-t\left(1 + \delta_{S_{\eta,m,n}}\right)/c_{LS}\right\}, \tag{8}$$

*where $D_0$ is a constant depending on the initialization, $\delta_{S_{\eta,m,n}} := \inf_{t>0} \frac{\mathcal{E}_{S_{\eta,m,n}}(\sqrt{\frac{d\nu_t}{d\pi}})}{\mathcal{E}(\sqrt{\frac{d\nu_t}{d\pi}})} - 1 \geq 0$ depends on $S_{\eta,m,n}$, $\mathcal{E}_{S_{\eta,m,n}}$ and $\mathcal{E}$ are the Dirichlet forms based on the swapping rate $S_{\eta,m,n}$ and are defined in (2), $c_{LS}$ is the constant of the log-Sobolev inequality for reLD without swaps.*

We detail the proof in Theorem.1 in Appendix B. Note that $S_{\eta,m,n} = 0$ leads to the same performance as the standard Langevin diffusion and $\delta_{S_{\eta,m,n}}$ is strictly positive when $\frac{d\nu_t}{d\pi}$ is asymmetric (Chen et al., 2019); given a smaller $\eta$ and $m$ or a large $n$, the variance becomes much reduced according to Lemma 1, yielding a much larger truncated swapping rate by Lemma 2 and a faster exponential convergence to the invariant measure $\pi$ compared to reSGLD.

Next, we estimate the upper bound of the 2-Wasserstein distance $\mathcal{W}(\mu_k, \nu_{k\eta})$, where $\mu_k$ denotes the probability measure associated with VR-reSGLD at iteration $k$. We first bypass the Grönwall inequality and conduct the change of measure to upper bound the relative entropy $D_{KL}(\mu_k|\nu_{k\eta})$ following (Raginsky et al., 2017). In addition to the approximation in the standard Langevin diffusion Raginsky et al. (2017), we also consider the change of Poisson measure following Yin & Zhu (2010); Gikhman & Skorokhod (1980) to handle the error from the stochastic swapping rate. We then extend the distance of relative entropy $D_{KL}(\mu_k|\nu_{k\eta})$ to the Wasserstein distance $\mathcal{W}_2(\mu_k, \nu_{k\eta})$ via a weighted transportation-cost inequality of Bolley & Villani (2005).

**Theorem 2 (Diffusion approximation)** *Assume the smoothness, the dissipativity and the gradient assumptions 1, 2 and 3 hold. Given a large enough batch size $n$, a small enough $m$ and $\eta$, we have*

$$\mathcal{W}_2(\mu_k, \nu_{k\eta}) \leq \mathcal{O}\left(dk^{3/2}\eta\left(\eta^{1/4} + \delta^{1/4} + \left(\frac{m^2}{n}\eta\right)^{1/8}\right)\right), \tag{9}$$

where $\delta$ is a constant that characterizes the scale of noise caused in mini-batch settings and the detail is given in Theorem 2 in Appendix C . Here the last term $\mathcal{O}\left(\left(\frac{m^2}{n}\eta\right)^{1/8}\right)$ comes from the error induced by the stochastic swapping rate, which disappears given a large enough batch size $n$ or a small enough update frequency $m$ and learning rate $\eta$. Note that our upper bound is linearly dependent on time approximately, which is much tighter than the exponential dependence using the Grönwall inequality. Admittedly, the result without swaps is slightly weaker than the diffusion approximation (3.1) in Raginsky et al. (2017) and we refer readers to Remark 3 in Appendix C.

Applying the triangle inequality for $\mathcal{W}_2(\mu_k, \nu_{k\eta})$ and $\mathcal{W}_2(\nu_{k\eta}, \pi)$ leads to the final result

**Theorem 3** *Assume the smoothness, the dissipativity and the gradient assumptions 1, 2 and 3 hold. Given a small enough learning rate $\eta$, update frequency $m$ and a large enough batch size $n$, we have*

$$\mathcal{W}_2(\mu_k, \pi) \leq \mathcal{O}\left(dk^{3/2}\eta\left(\eta^{1/4} + \delta^{1/4} + \left(\frac{m^2}{n}\eta\right)^{1/8}\right)\right) + \mathcal{O}\left(e^{\frac{-k\eta(1+\delta_{S_{\eta,m,n}})}{c_{LS}}}\right).$$

This theorem implies that increasing the batch size $n$ or decreasing the update frequency $m$ not only reduces the numerical error but also potentially leads to a faster exponential convergence of the continuous-time dynamics via a much larger swapping rate $S_{\eta,m,n}$.

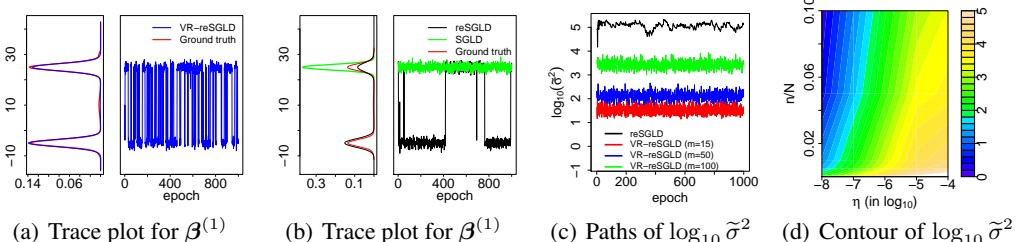

(a) Trace plot for $\boldsymbol{\beta}^{(1)}$    (b) Trace plot for $\boldsymbol{\beta}^{(1)}$    (c) Paths of $\log_{10} \tilde{\sigma}^2$    (d) Contour of $\log_{10} \tilde{\sigma}^2$

Figure 2: Trace plots, KDEs of $\boldsymbol{\beta}^{(1)}$, and sensitivity study of $\tilde{\sigma}^2$ with respect to $m, \eta$ and $n$.

## 5   EXPERIMENTS

### 5.1   SIMULATIONS OF GAUSSIAN MIXTURE DISTRIBUTIONS

We first study the proposed variance-reduced replica exchange stochastic gradient Langevin dynamics algorithm (VR-reSGLD) on a Gaussian mixture distribution (Dubey et al., 2016). The distribution follows from $x_i | \beta \sim 0.5 \mathrm{N}(\beta, \sigma^2) + 0.5 \mathrm{N}(\phi - \beta, \sigma^2)$, where $\phi = 20$, $\sigma = 5$ and $\beta = -5$. We use a training dataset of size $N = 10^5$ and propose to estimate the posterior distribution over $\beta$. We compare the performance of VR-reSGLD against that of the standard stochastic gradient Langevin dynamics (SGLD), and replica exchange SGLD (reSGLD).

In Figs 2(a) and 2(b), we present trace plots and kernel density estimates (KDE) of samples generated from VR-reSGLD with $m = 40$, $\tau^{(1)} = 10$ [†], $\tau^{(2)} = 1000$, $\eta = 1e - 7$, and $F = 1$; reSGLD adopt the same hyper-parameters except for $F = 100$ because a smaller $F$ may fail to propose any swaps; SGLD uses $\eta = 1e - 7$ and $\tau = 10$. As the posterior density is intractable, we consider a ground truth by running replica exchange Langevin dynamics with long enough iterations. We observe that VR-reSGLD is able to fully recover the posterior density, and successfully jump between the two modes passing the energy barrier frequently enough. By contrast, SGLD, initialized at $\beta_0 = 30$, is attracted to the nearest mode and fails to escape throughout the run; reSGLD manages to jump between the two modes, however, $F$ is chosen as large as 100, which induces a large bias and only yields three to five swaps and exhibits the metastability issue. In Figure 2(c), we present the evolution of the variance for VR-reSGLD over a range of different $m$ and compare it with reSGLD. We see that the variance reduction mechanism has successfully reduced the variance by hundreds of times. In Fig 2(d), we present the sensitivity study of $\tilde{\sigma}^2$ as a function of the ratio $n/N$ and the learning rate $\eta$; for this estimate we average out 10 realizations of VR-reSGLD, and our results agree with the theoretical analysis in Lemma 1.

### 5.2   NON-CONVEX OPTIMIZATION FOR IMAGE DATA

We further test the proposed algorithm on CIFAR10 and CIFAR100. We choose the 20, 32, 56-layer residual networks as the training models and denote them by ResNet-20, ResNet-32, and ResNet-56, respectively. Considering the wide adoption of M-SGD, stochastic gradient Hamiltonian Monte Carlo (SGHMC) is selected as the baseline. We refer to the standard replica exchange SGHMC algorithm as reSGHMC and the variance-reduced reSGHMC algorithm as VR-reSGHMC. We also include another baseline called cyclical stochastic gradient MCMC (cycSGHMC), which proposes a cyclical learning rate schedule. To make a fair comparison, we test the variance-reduced replica exchange SGHMC algorithm with cyclic learning rates and refer to it as cVR-reSGHMC.

We run M-SGD, SGHMC and (VR-)reSGHMC for 500 epochs. For these algorithms, we follow a setup from Deng et al. (2020). We fix the learning rate $\eta_k^{(1)} = 2e\text{-}6$ in the first 200 epochs and decay it by 0.984 afterwards. For SGHMC and the low-temperature processes of (VR-)reSGHMC, we anneal the temperature following $\tau_k^{(1)} = 0.01/1.02^k$ in the beginning and keep it fixed after the burn-in steps; regarding the high-temperature process, we set $\eta_k^{(2)} = 1.5\eta_k^{(1)}$ and $\tau_k^{(2)} = 5\tau_k^{(1)}$. The initial correction factor $F_0$ is fixed at $1.5e5$. The thinning factor $\mathbb{T}$ is set to 256. In particular for

---

[†]We choose $\tau^{(1)} = 10$ instead of 1 to avoid peaky modes for ease of illustration.

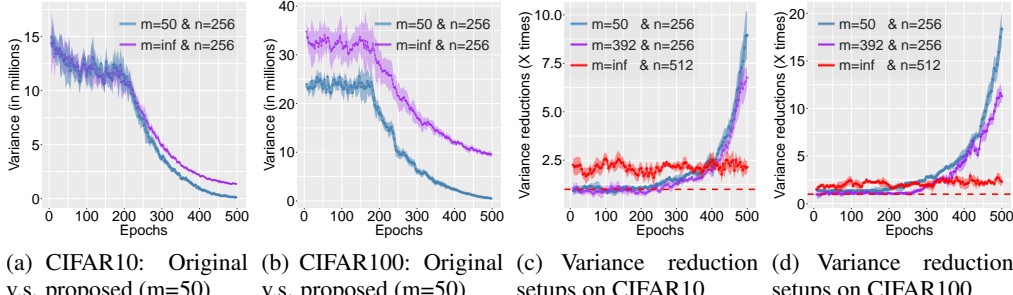

(a) CIFAR10: Original v.s. proposed (m=50) (b) CIFAR100: Original v.s. proposed (m=50) (c) Variance reduction setups on CIFAR10 (d) Variance reduction setups on CIFAR100

Figure 3: Variance reduction on the noisy energy estimators on CIFAR10 & CIFAR100 datasets.

cycSGHMC, we run the algorithm for 1000 epochs and choose the cosine learning rate schedule with 5 cycles; $\eta_0$ is set to 1e-5; we fix the temperature 0.001 and the threshold 0.7 for collecting the samples. Similarly, we propose the cosine learning rate for cVR-reSGHMC with 2 cycles and run it for 500 epochs using the same temperature 0.001. We only study the low-temperature process for the replica exchange algorithms. Each experiment is repeated five times to obtain the mean and 2 standard deviations.

We evaluate the performance of variance reduction using VR-reSGHMC and compare it with reS-GHMC. We first increase the batch size $n$ from 256 to 512 for reSGHMC and notice that the reduction of variance is around 2 times (see the red curves in Fig.3(c,d)). Next, we try $m = 50$ and $n = 256$ for the VR-reSGHMC algorithm, which updates the control variates every 50 iterations. As shown in Fig.3(a,b), during the first 200 epochs, where the largest learning rate is used, the variance of VR-reSGHMC is slightly reduced by 37% on CIFAR100 and doesn't make a difference on CIFAR10. However, as the learning rate and the temperature decrease, the reduction of the variance gets more significant. We see from Fig.3(c,d) that the reduction of variance can be *up to 10 times on CIFAR10 and 20 times on CIFAR100*. This is consistent with our theory proposed in Lemma 1. The reduction of variance based on VR-reSGHMC starts to outperform the baseline with $n = 512$ when the epoch is higher than 370 on CIFAR10 and 250 on CIFAR100. We also try $m = 392$, which updates the control variates every 2 epochs, and find a similar pattern.

For computational reasons, we choose $m = 392$ and $n = 256$ for (c)VR-reSGHMC and compare them with the baseline algorithms. With the help of swaps between two SGHMC chains, reSGHMC already obtains remarkable performance (Deng et al., 2020) and five swaps often lead to an optimal performance. However, VR-reSGHMC still outperforms reSGHMC by around 0.2% on CIFAR10 and 1% improvement on CIFAR100 (Table.1) and *the number of swaps is increased to around a hundred under the same setting*. We also try cyclic learning rates and compare cVR-reSGHMC with cycSGHMC, we see cVR-reSGHMC outperforms cycSGHMC significantly even if cycSGHMC is running 1000 epochs, which may be more costly than cVR-reSGHMC due to the lack of mechanism in parallelism. Note that cVR-reSGHMC keeps the temperature the same instead of annealing it as in VR-reSGHMC, which is more suitable for uncertainty quantification.

TABLE 1: PREDICTION ACCURACIES (%) BASED ON BAYESIAN MODEL AVERAGING. IN PAR-TICULAR, M-SGD AND SGHMC RUN 500 EPOCHS USING A SINGLE CHAIN; CYCSGHMC RUN 1000 EPOCHS USING A SINGLE CHAIN; REPLICA EXCHANGE ALGORITHMS RUN 500 EPOCHS USING TWO CHAINS WITH DIFFERENT TEMPERATURES.

| METHOD | CIFAR10 | | | CIFAR100 | | |
|---|---|---|---|---|---|---|
| | RESNET20 | RESNET32 | RESNET56 | RESNET20 | RESNET32 | RESNET56 |
| M-SGD | 94.07±0.11 | 95.11±0.07 | 96.05±0.21 | 71.93±0.13 | 74.65±0.20 | 78.76±0.24 |
| SGHMC | 94.16±0.13 | 95.17±0.08 | 96.04±0.18 | 72.09±0.14 | 74.80±0.19 | 78.95±0.22 |
| reSGHMC | 94.56±0.23 | 95.44±0.16 | 96.15±0.17 | 73.94±0.34 | 76.38±0.23 | 79.86±0.26 |
| VR-reSGHMC | **94.84±0.11** | **95.62±0.09** | **96.32±0.15** | **74.83±0.18** | **77.40±0.27** | **80.62±0.22** |
| cycSGHMC | 94.61±0.15 | 95.56±0.12 | 96.19±0.17 | 74.21±0.22 | 76.60±0.25 | 80.39±0.21 |
| cVR-reSGHMC | **94.91±0.10** | **95.64±0.13** | **96.36±0.16** | **75.02±0.19** | **77.58±0.21** | **80.50±0.25** |

Regarding the training cost and the treatment for improving the performance of variance reduction using adaptive coefficients in the early period, we refer interested readers to Appendix E.

For the detailed implementations, we release the code at `https://github.com/WayneDW/Variance_Reduced_Replica_Exchange_Stochastic_Gradient_MCMC`.

### 5.3 UNCERTAINTY QUANTIFICATION FOR UNKNOWN SAMPLES

A reliable model not only makes the right decision among potential candidates but also casts doubts on irrelevant choices. For the latter, we follow Lakshminarayanan et al. (2017) and evaluate the uncertainty on out-of-distribution samples from unseen classes. To avoid over-confident predictions on unknown classes, the ideal predictions should yield a higher uncertainty on the out-of-distribution samples, while maintaining the accurate uncertainty for the in-distribution samples.

Continuing the setup in Sec.5.2, we collect the ResNet20 models trained on CIFAR10 and quantify the entropy on the Street View House Numbers (SVHN) dataset, which contains 26,032 RGB testing images of digits instead of objects. We compare cVR-reSGHMC with M-SGD, SGHMC, reSGHMC, and cSGHMC. Ideally, the predictive distribution should be the uniform distribution and leads to the highest entropy. We present the empirical cumulative distribution function (CDF) of the entropy of the predictions on SVHN and report it in Fig.4. As shown in the left figure,

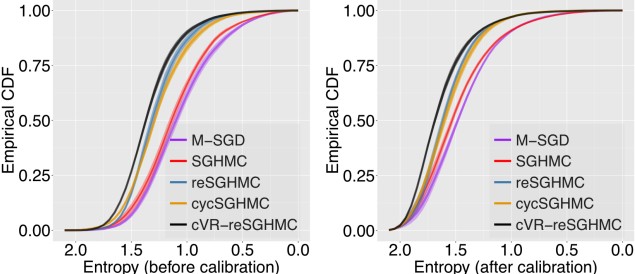

Figure 4: CDF of entropy for predictions on SVHN via CIFAR10 models. A temperature scaling is used in calibrations.

M-SGD shows the smallest probability for high-entropy predictions, implying the weakness of stochastic optimization methods in uncertainty estimates. By contrast, the proposed cVR-reSGHMC yields the highest probability for predictions of high entropy. Admittedly, the standard ResNet models are poorly calibrated in the predictive probabilities and lead to inaccurate confidence. To alleviate this issue, we adopt the temperature-scaling method with a scale of 2 to calibrate the predictive distribution (Guo et al., 2017) and present the entropy in Fig.4 (right). In particular, we see that 77% of the predictions from cVR-reSGHMC yields the entropy higher than 1.5, which is 7% higher than reSGHMC and 10% higher than cSGHMC and much better than the others.

For more discussions of uncertainty estimates on both datasets, we leave the results in Appendix F.

## 6 CONCLUSION

We propose the variance-reduced replica exchange stochastic gradient Langevin dynamics algorithm to accelerate the convergence by reducing the variance of the noisy energy estimators. Theoretically, this is *the first variance reduction method that yields the potential of exponential accelerations* instead of solely reducing the discretization error. In addition, we bypass the Grönwall inequality to avoid the crude numerical error and consider a change of Poisson measure in the generalized Girsanov theorem to obtain a much tighter upper bound. Since our variance reduction only conducts on the noisy energy estimators and is not applied to the noisy gradients, the standard hyper-parameter setting can be also naturally imported, which greatly facilitates the training of deep neural works.

## ACKNOWLEDGMENT

We would like to thank Maxim Raginsky and the anonymous reviewers for their insightful suggestions. Liang's research was supported in part by the grants DMS-2015498, R01-GM117597 and R01-GM126089. Lin acknowledges the support from NSF (DMS-1555072, DMS-1736364), BNL Subcontract 382247, W911NF-15-1-0562, and DE-SC0021142.

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

## A    PRELIMINARIES

**Notation** We denote the deterministic energy based on the parameter $\boldsymbol{\beta}$ by $L(\boldsymbol{\beta}) = \sum_{i=1}^{N} L(\mathbf{x}_i|\boldsymbol{\beta})$ using the full dataset of size $N$. We denote the unbiased stochastic energy estimator by $\frac{N}{n} \sum_{i\in B} L(\mathbf{x}_i|\boldsymbol{\beta})$ using the mini-batch of data $B$ of size $n$. The same style of notations is also applicable to the gradient for consistency. We denote the Euclidean $L^2$ norm by $\|\cdot\|$. To prove the desired results, we need the following assumptions:

**Assumption 1 (Smoothness)** *The energy function $L(\mathbf{x}_i|\cdot)$ is $C_N$-smoothness if there exists a constant $C_N > 0$ such that $\forall \boldsymbol{\beta}_1, \boldsymbol{\beta}_2 \in \mathbb{R}^d$, $i \in \{1, 2, \cdots, N\}$, we have*

$$\|\nabla L(\mathbf{x}_i|\boldsymbol{\beta}_1) - \nabla L(\mathbf{x}_i|\boldsymbol{\beta}_2)\| \leq C_N \|\boldsymbol{\beta}_1 - \boldsymbol{\beta}_2\|. \tag{10}$$

*Note that the above condition further implies for a constant $C = NC_N$ and $\forall \boldsymbol{\beta}_1, \boldsymbol{\beta}_2 \in \mathbb{R}^d$, we have*

$$\|\nabla L(\boldsymbol{\beta}_1) - \nabla L(\boldsymbol{\beta}_2)\| \leq C \|\boldsymbol{\beta}_1 - \boldsymbol{\beta}_2\|. \tag{11}$$

The smoothness conditions (10) and (11) are standard tools in studying the convergence of SGLD in (Xu et al., 2018) and Raginsky et al. (2017), respectively.

**Assumption 2 (Dissipativity)** *The energy function $L(\cdot)$ is $(a, b)$-dissipative if there exist constants $a > 0$ and $b \geq 0$ such that $\forall \boldsymbol{\beta} \in \mathbb{R}^d$, $\langle \boldsymbol{\beta}, \nabla L(\boldsymbol{\beta}) \rangle \geq a\|\boldsymbol{\beta}\|^2 - b$.*

The dissipativity condition implies that the Markov process is able to move inward on average regardless of the starting position. It has been widely used in proving the geometric ergodicity of dynamic systems (Mattingly et al., 2002; Raginsky et al., 2017; Xu et al., 2018).

**Assumption 3 (Gradient oracle)** *There exists a constant $\delta \in [0, 1)$ such that for any $\boldsymbol{\beta}$, we have*

$$\mathbb{E}[\|\nabla \widetilde{L}(\boldsymbol{\beta}) - \nabla L(\boldsymbol{\beta})\|^2] \leq 2\delta(C^2 \|\boldsymbol{\beta}\|^2 + \Phi^2), \tag{12}$$

*where $\Phi$ is a positive constant. The same assumption has been used in Raginsky et al. (2017) to control the stochastic noise from the gradient.*

## B    EXPONENTIAL ACCELERATIONS VIA VARIANCE REDUCTION

We aim to build an efficient estimator to approximate the deterministic swapping rate $S(\boldsymbol{\beta}^{(1)}, \boldsymbol{\beta}^{(2)})$

$$S(\boldsymbol{\beta}^{(1)}, \boldsymbol{\beta}^{(2)}) = e^{\left(\frac{1}{\tau^{(1)}} - \frac{1}{\tau^{(2)}}\right)\left(\sum_{i=1}^{N} L(\mathbf{x}_i|\boldsymbol{\beta}^{(1)}) - \sum_{i=1}^{N} L(\mathbf{x}_i|\boldsymbol{\beta}^{(2)})\right)}. \tag{13}$$

In big data problems and deep learning, it is too expensive to evaluate the energy $\sum_{i=1}^{N} L(\mathbf{x}_i|\boldsymbol{\beta})$ for each $\boldsymbol{\beta}$ for a large $N$. To handle the computational issues, a popular solution is to use the unbiased stochastic energy $\frac{N}{n} \sum_{i\in B} L(\mathbf{x}_i|\boldsymbol{\beta})$ for a random mini-batch data $B$ of size $n$. However, a näive replacement of $\sum_{i=1}^{N} L(\mathbf{x}_i|\boldsymbol{\beta})$ by $\frac{N}{n} \sum_{i\in B} L(\mathbf{x}_i|\boldsymbol{\beta})$ leads to a large bias to the swapping rate. To remove such a bias, we follow Deng et al. (2020) and consider the corrected swapping rate

$$\widehat{S}(\boldsymbol{\beta}^{(1)}, \boldsymbol{\beta}^{(2)}) = e^{\left(\frac{1}{\tau^{(1)}} - \frac{1}{\tau^{(2)}}\right)\left(\frac{N}{n} \sum_{i\in B} L(\mathbf{x}_i|\boldsymbol{\beta}^{(1)}) - \frac{N}{n} \sum_{i\in B} L(\mathbf{x}_i|\boldsymbol{\beta}^{(2)}) - \left(\frac{1}{\tau^{(1)}} - \frac{1}{\tau^{(2)}}\right)\frac{\widehat{\sigma}^2}{2}\right)}, \tag{14}$$

where $\widehat{\sigma}^2$ denotes the variance of $\frac{N}{n} \sum_{i\in B} L(\mathbf{x}_i|\boldsymbol{\beta}^{(1)}) - \frac{N}{n} \sum_{i\in B} L(\mathbf{x}_i|\boldsymbol{\beta}^{(2)})$. [*] Empirically, $\widehat{\sigma}^2$ is quite large, resulting in almost no swaps and insignificant accelerations. To propose more effective swaps, we consider the variance-reduced estimator

$$\widetilde{L}(B_k|\boldsymbol{\beta}_k) = \frac{N}{n} \sum_{i\in B_k} \left( L(\mathbf{x}_i|\boldsymbol{\beta}_k) - L\left(\mathbf{x}_i\Big|\boldsymbol{\beta}_{m\lfloor\frac{k}{m}\rfloor}\right) \right) + \sum_{i=1}^{N} L\left(\mathbf{x}_i\Big|\boldsymbol{\beta}_{m\lfloor\frac{k}{m}\rfloor}\right), \tag{15}$$

where the control variate $\boldsymbol{\beta}_{m\lfloor\frac{k}{m}\rfloor}$ is updated every $m$ iterations. Denote the variance of $\widetilde{L}(B|\boldsymbol{\beta}^{(1)}) - \widetilde{L}(B|\boldsymbol{\beta}^{(2)})$ by $\widetilde{\sigma}^2$. The variance-reduced stochastic swapping rate follows

$$\widetilde{S}_{\eta,m,n}(\boldsymbol{\beta}^{(1)}, \boldsymbol{\beta}^{(2)}) = e^{\left(\frac{1}{\tau^{(1)}} - \frac{1}{\tau^{(2)}}\right)\left(\widetilde{L}(B|\boldsymbol{\beta}^{(1)}) - \widetilde{L}(B|\boldsymbol{\beta}^{(2)}) - \left(\frac{1}{\tau^{(1)}} - \frac{1}{\tau^{(2)}}\right)\frac{\widetilde{\sigma}^2}{2}\right)}. \tag{16}$$

---

[*]We only consider the case of $F = 1$ in the stochastic swapping rate for ease of analysis.

Using the strategy of variance reduction, we can lay down the first result, which differs from the existing variance reduction methods in that we only conduct variance reduction in the energy estimator for the class of SGLD algorithms.

**Lemma B1 (Variance-reduced energy estimator)** *Under the smoothness and dissipativity assumptions 1 and 2, the variance of the variance-reduced energy estimator $\widetilde{L}(B_k|\boldsymbol{\beta}_k^{(h)})$, where $h \in \{1, 2\}$, is upper bounded by*

$$\mathrm{Var}\left(\widetilde{L}(B_k|\boldsymbol{\beta}_k^{(h)})\right) \leq \frac{m^2\eta}{n}D_R^2\left(\frac{2\eta}{n}(2C^2\Psi_{d,\tau^{(2)},C,a,b} + 2Q^2) + 4\tau^{(2)}d\right). \tag{17}$$

*where $D_R = CR + \max_{i \in \{1,2,\cdots,N\}} N\|\nabla L(\mathbf{x}_i|\boldsymbol{\beta}_\star)\| + \frac{Cb}{a}$ and $R$ is the radius of a sufficiently large ball that contains $\boldsymbol{\beta}_k^{(h)}$ for $h \in \{1, 2\}$.*

**Proof**

$$
\begin{aligned}
&\mathrm{Var}\left(\widetilde{L}(B_k|\boldsymbol{\beta}_k^{(h)})\right)\\
=&\mathbb{E}\left[\left(\frac{N}{n}\sum_{i\in B_k}\left[L(\mathbf{x}_i|\boldsymbol{\beta}_k^{(h)}) - L\left(\mathbf{x}_i\Big|\boldsymbol{\beta}_{m\lfloor\frac{k}{m}\rfloor}^{(h)}\right)\right] + \sum_{j=1}^{N}L\left(\mathbf{x}_j\Big|\boldsymbol{\beta}_{m\lfloor\frac{k}{m}\rfloor}^{(h)}\right) - \sum_{j=1}^{N}L(\mathbf{x}_j|\boldsymbol{\beta}_k^{(h)})\right)^2\right]\\
=&\mathbb{E}\left[\left(\frac{N}{n}\sum_{i\in B_k}\left[L(\mathbf{x}_i|\boldsymbol{\beta}_k^{(h)}) - L\left(\mathbf{x}_i\Big|\boldsymbol{\beta}_{m\lfloor\frac{k}{m}\rfloor}^{(h)}\right) + \frac{1}{N}\left(\sum_{j=1}^{N}L\left(\mathbf{x}_j\Big|\boldsymbol{\beta}_{m\lfloor\frac{k}{m}\rfloor}^{(h)}\right) - \sum_{j=1}^{N}L(\mathbf{x}_j|\boldsymbol{\beta}_k^{(h)})\right)\right]\right)^2\right]\\
=&\frac{N^2}{n^2}\mathbb{E}\left[\left(\sum_{i\in B_k}\left[L(\mathbf{x}_i|\boldsymbol{\beta}_k^{(h)}) - L\left(\mathbf{x}_i\Big|\boldsymbol{\beta}_{m\lfloor\frac{k}{m}\rfloor}^{(h)}\right) + \frac{1}{N}\left(\sum_{j=1}^{N}L\left(\mathbf{x}_j\Big|\boldsymbol{\beta}_{m\lfloor\frac{k}{m}\rfloor}^{(h)}\right) - \sum_{j=1}^{N}L(\mathbf{x}_j|\boldsymbol{\beta}_k^{(h)})\right)\right]\right)^2\right]\\
=&\frac{N^2}{n^2}\sum_{i\in B_k}\mathbb{E}\left[\left(L(\mathbf{x}_i|\boldsymbol{\beta}_k^{(h)}) - L\left(\mathbf{x}_i\Big|\boldsymbol{\beta}_{m\lfloor\frac{k}{m}\rfloor}^{(h)}\right) - \frac{1}{N}\left[\sum_{j=1}^{N}L(\mathbf{x}_j|\boldsymbol{\beta}_k^{(h)}) - \sum_{j=1}^{N}L\left(\mathbf{x}_j\Big|\boldsymbol{\beta}_{m\lfloor\frac{k}{m}\rfloor}^{(h)}\right)\right]\right)^2\right]\\
\leq&\frac{N^2}{n^2}\sum_{i\in B_k}\mathbb{E}\left[\left(L(\mathbf{x}_i|\boldsymbol{\beta}_k^{(h)}) - L\left(\mathbf{x}_i\Big|\boldsymbol{\beta}_{m\lfloor\frac{k}{m}\rfloor}^{(h)}\right)\right)^2\right]\\
\leq&\frac{D_R^2}{n}\mathbb{E}\left[\left\|\boldsymbol{\beta}_k^{(h)} - \boldsymbol{\beta}_{m\lfloor\frac{k}{m}\rfloor}^{(h)}\right\|^2\right],
\end{aligned}
\tag{18}
$$

where the last equality follows from the fact that $\mathbb{E}[(\sum_{i=1}^{n}x_i)^2] = \sum_{i=1}^{n}\mathbb{E}[x_i^2]$ for independent variables $\{x_i\}_{i=1}^{n}$ with mean 0. The first inequality follows from $\mathbb{E}[(x - \mathbb{E}[x])^2] \leq \mathbb{E}[x^2]$ and the last inequality follows from Lemma D1, where $D_R = CR + \max_{i \in \{1,2,\cdots,N\}} N\|\nabla L(\mathbf{x}_i|\boldsymbol{\beta}_\star)\| + \frac{Cb}{a}$ and $R$ is the radius of a sufficiently large ball that contains $\boldsymbol{\beta}_k^{(h)}$ for $h \in \{1, 2\}$.

Next, we bound $\mathbb{E}\left[\left\|\boldsymbol{\beta}_k^{(h)} - \boldsymbol{\beta}_{m\lfloor\frac{k}{m}\rfloor}^{(h)}\right\|^2\right]$ as follows

$$\mathbb{E}\left[\left\|\boldsymbol{\beta}_k^{(h)} - \boldsymbol{\beta}_{m\lfloor\frac{k}{m}\rfloor}^{(h)}\right\|^2\right] \leq \mathbb{E}\left[\left\|\sum_{j=m\lfloor\frac{k}{m}\rfloor}^{k-1}(\boldsymbol{\beta}_{j+1}^{(h)} - \boldsymbol{\beta}_j^{(h)})\right\|^2\right] \leq m\sum_{j=m\lfloor\frac{k}{m}\rfloor}^{k-1}\mathbb{E}\left[\left\|(\boldsymbol{\beta}_{j+1}^{(h)} - \boldsymbol{\beta}_j^{(h)})\right\|^2\right]. \tag{19}$$

For each term, we have the following bound

$$
\begin{aligned}
\mathbb{E}\left[\left\|\boldsymbol{\beta}_{j+1}^{(h)} - \boldsymbol{\beta}_j^{(h)}\right\|^2\right] =& \mathbb{E}\left[\left\|\eta\frac{N}{n}\sum_{i\in B_k}\nabla L(\mathbf{x}_i|\boldsymbol{\beta}_k^{(h)}) + \sqrt{2\eta\tau^{(h)}}\boldsymbol{\xi}_k\right\|^2\right] \\
\leq& \frac{2\eta^2 N^2}{n^2}\sum_{i\in B_k}\mathbb{E}\left[\left\|\nabla L(\mathbf{x}_i|\boldsymbol{\beta}_k^{(h)})\right\|^2\right] + 4\eta\tau^{(2)}d \\
\leq& \frac{2\eta^2}{n}(2C^2\mathbb{E}[\|\boldsymbol{\beta}_k^{(h)}\|^2] + 2Q^2) + 4\eta\tau^{(2)}d \\
\leq& \frac{2\eta^2}{n}(2C^2\Psi_{d,\tau^{(2)},C,a,b} + 2Q^2) + 4\eta\tau^{(2)}d,
\end{aligned}
\tag{20}
$$

where the first inequality follows by $\mathbb{E}[\|a+b\|^2] \leq 2\mathbb{E}[\|a\|^2] + 2\mathbb{E}[\|b\|^2]$, the i.i.d of the data points and $\tau^{(1)} \leq \tau^{(2)}$ for $h \in \{1, 2\}$; the second inequality follows by Lemma D2; the last inequality follows from Lemma D3.

Combining (18), (19) and (20), we have

$$
\mathrm{Var}\left(\widetilde{L}(B_k|\boldsymbol{\beta}_k^{(h)})\right) \leq \frac{m^2\eta}{n}D_R^2\left(\frac{2\eta}{n}(2C^2\Psi_{d,\tau^{(2)},C,a,b} + 2Q^2) + 4\tau^{(2)}d\right). \tag{21}
$$

∎

Since $\mathrm{Var}\left(\widetilde{L}(B_k|\boldsymbol{\beta}_k^{(h)})\right) \leq \mathrm{Var}\left(\frac{N}{n}\sum_{i\in B}L(\mathbf{x}_i|\boldsymbol{\beta}_k)\right) + \mathrm{Var}\left(\frac{N}{n}\sum_{i\in B}L\left(\mathbf{x}_i\Big|\boldsymbol{\beta}_{m\lfloor\frac{k}{m}\rfloor}\right)\right)$ by definition, $\mathrm{Var}\left(\widetilde{L}(B_k|\boldsymbol{\beta}_k^{(h)})\right)$ is upper bounded by $\mathcal{O}\left(\min\{\widehat{\sigma}^2, \frac{m^2\eta}{n}\}\right)$, which becomes much smaller using a small learning rate $\eta$, a shorter period $m$ and a large batch size $n$.

Note that $\widetilde{S}_{\eta,m,n}(\boldsymbol{\beta}^{(1)}, \boldsymbol{\beta}^{(2)})$ is defined on the unbounded support $[0, \infty]$ and $\mathbb{E}[\widetilde{S}_{\eta,m,n}(\boldsymbol{\beta}^{(1)}, \boldsymbol{\beta}^{(2)})] = S(\boldsymbol{\beta}^{(1)}, \boldsymbol{\beta}^{(2)})$ regardless of the scale of $\widetilde{\sigma}^2$. To satisfy the (stochastic) reversibility condition, we consider the truncated swapping rate $\min\{1, \widetilde{S}_{\eta,m,n}(\boldsymbol{\beta}^{(1)}, \boldsymbol{\beta}^{(2)})\}$, which still targets the same invariant distribution (see section 3.1 (Quiroz et al., 2019) for details). We can show that the swapping rate may even decrease exponentially as the variance increases.

**Lemma B2 (Variance reduction for larger swapping rates)** *Given a large enough batch size $n$, the variance-reduced energy estimator $\widetilde{L}(B_k|\boldsymbol{\beta}_k^{(h)})$ yields a truncated swapping rate that satisfies*

$$
\mathbb{E}[\min\{1, \widetilde{S}_{\eta,m,n}(\boldsymbol{\beta}^{(1)}, \boldsymbol{\beta}^{(2)})\}] \approx \min\left\{1, S(\boldsymbol{\beta}^{(1)}, \boldsymbol{\beta}^{(2)})\left(\mathcal{O}\left(\frac{1}{n^2}\right) + e^{-\mathcal{O}\left(\frac{m^2\eta}{n} + \frac{1}{n^2}\right)}\right)\right\}. \tag{22}
$$

**Proof**

By central limit theorem, the energy estimator $\frac{N}{n}\sum_{i\in B}L(\mathbf{x}_i|\boldsymbol{\beta}_k)$ converges in distribution to a normal distributions as the batch size $n$ goes to infinity. In what follows, the variance-reduced estimator $\widetilde{L}(B_k|\boldsymbol{\beta}_k)$ also converges to a normal distribution, where the corresponding estimator is denoted by $\widetilde{\mathbb{L}}(B_k|\boldsymbol{\beta}_k)$. Now the swapping rate $\mathbb{S}_{\eta,m,n}(\cdot, \cdot)$ based on normal estimators follows

$$
\mathbb{S}_{\eta,m,n}(\boldsymbol{\beta}^{(1)}, \boldsymbol{\beta}^{(2)}) = e^{\left(\frac{1}{\tau^{(1)}} - \frac{1}{\tau^{(2)}}\right)\left(\widetilde{\mathbb{L}}(B|\boldsymbol{\beta}^{(1)}) - \widetilde{\mathbb{L}}(B|\boldsymbol{\beta}^{(2)}) - \left(\frac{1}{\tau^{(1)}} - \frac{1}{\tau^{(2)}}\right)\frac{\bar{\sigma}^2}{2}\right)}, \tag{23}
$$

where $\bar{\sigma}^2$ denotes the variance of $\widetilde{\mathbb{L}}(B|\boldsymbol{\beta}^{(1)}) - \widetilde{\mathbb{L}}(B|\boldsymbol{\beta}^{(2)})$. Note that $\mathbb{S}_{\eta,m,n}(\boldsymbol{\beta}^{(1)}, \boldsymbol{\beta}^{(2)})$ follows a log-normal distribution with mean $\log S(\boldsymbol{\beta}^{(1)}, \boldsymbol{\beta}^{(2)}) - \left(\frac{1}{\tau^{(1)}} - \frac{1}{\tau^{(2)}}\right)^2\frac{\bar{\sigma}^2}{2}$ and variance $\left(\frac{1}{\tau^{(1)}} - \frac{1}{\tau^{(2)}}\right)^2\bar{\sigma}^2$ on the log-scale, and $S(\boldsymbol{\beta}^{(1)}, \boldsymbol{\beta}^{(2)})$ is the deterministic swapping rate defined in (13). Applying Lemma D4, we have

$$
\mathbb{E}[\min\{1, \mathbb{S}_{\eta,m,n}(\boldsymbol{\beta}^{(1)}, \boldsymbol{\beta}^{(2)})\}] = \mathcal{O}\left(S(\boldsymbol{\beta}^{(1)}, \boldsymbol{\beta}^{(2)})\exp\left\{-\frac{\left(\frac{1}{\tau^{(1)}} - \frac{1}{\tau^{(2)}}\right)^2\bar{\sigma}^2}{8}\right\}\right). \tag{24}
$$

Moreover, $\bar{\sigma}^2$ differs from $\widetilde{\sigma}^2$, the variance of $\widetilde{L}(B|\boldsymbol{\beta}^{(1)}) - \widetilde{L}(B|\boldsymbol{\beta}^{(2)})$, by at most a bias of $\mathcal{O}(\frac{1}{n^2})$ according to the estimate of the third term of (S2) in Quiroz et al. (2019) and $\widetilde{\sigma}^2 \leq$

$\text{Var}\left(\widetilde{L}(B_k|\boldsymbol{\beta}_k^{(1)})\right) + \text{Var}\left(\widetilde{L}(B_k|\boldsymbol{\beta}_k^{(2)})\right)$, where both $\text{Var}\left(\widetilde{L}(B_k|\boldsymbol{\beta}_k^{(1)})\right)$ and $\text{Var}\left(\widetilde{L}(B_k|\boldsymbol{\beta}_k^{(2)})\right)$ are upper bounded by $\frac{m^2\eta}{n}D_R^2\left(\frac{2\eta}{n}(2C^2\Psi_{d,\tau^{(2)},C,a,b} + 2Q^2) + 4\tau d\right)$ by Lemma B1, it follows that

$$\mathbb{E}[\min\{1, \mathbb{S}_{\eta,m,n}(\boldsymbol{\beta}^{(1)},\boldsymbol{\beta}^{(2)})\}] \leq S(\boldsymbol{\beta}^{(1)},\boldsymbol{\beta}^{(2)})e^{-\mathcal{O}\left(\frac{m^2\eta}{n} + \frac{1}{n^2}\right)}. \tag{25}$$

Applying $\min\{1, \mathbb{A} + \mathbb{B}\} \leq \min\{1, \mathbb{A}\} + |\mathbb{B}|$, we have

$$\begin{aligned}
&\mathbb{E}[\min\{1, \widetilde{S}_{\eta,m,n}(\boldsymbol{\beta}^{(1)},\boldsymbol{\beta}^{(2)})\}] \\
=&\mathbb{E}\big[\min\big\{1, \underbrace{\widetilde{S}_{\eta,m,n}(\boldsymbol{\beta}^{(1)},\boldsymbol{\beta}^{(2)}) - \mathbb{S}_{\eta,m,n}(\boldsymbol{\beta}^{(1)},\boldsymbol{\beta}^{(2)})}_{\mathbb{B}} + \underbrace{\mathbb{S}_{\eta,m,n}(\boldsymbol{\beta}^{(1)},\boldsymbol{\beta}^{(2)})}_{\mathbb{A}}\big\}\big] \\
\leq&\underbrace{\mathbb{E}\left[\left|\widetilde{S}_{\eta,m,n}(\boldsymbol{\beta}^{(1)},\boldsymbol{\beta}^{(2)}) - \mathbb{S}_{\eta,m,n}(\boldsymbol{\beta}^{(1)},\boldsymbol{\beta}^{(2)})\right|\right]}_{\mathcal{I}} + \underbrace{\mathbb{E}[\min\{1, \mathbb{S}_{\eta,m,n}(\boldsymbol{\beta}^{(1)},\boldsymbol{\beta}^{(2)})\}]}_{\text{see formula (25)}}
\end{aligned} \tag{26}$$

By the triangle inequality, we can further upper bound the first term $\mathcal{I}$

$$\begin{aligned}
&\mathbb{E}\left[\left|\widetilde{S}_{\eta,m,n}(\boldsymbol{\beta}^{(1)},\boldsymbol{\beta}^{(2)}) - \mathbb{S}_{\eta,m,n}(\boldsymbol{\beta}^{(1)},\boldsymbol{\beta}^{(2)})\right|\right] \\
\leq&\underbrace{\left|\mathbb{E}[\widetilde{S}_{\eta,m,n}(\boldsymbol{\beta}^{(1)},\boldsymbol{\beta}^{(2)})] - S(\boldsymbol{\beta}^{(1)},\boldsymbol{\beta}^{(2)})\right|}_{\mathcal{I}_1} + \underbrace{\left|S(\boldsymbol{\beta}^{(1)},\boldsymbol{\beta}^{(2)}) - \mathbb{E}[\mathbb{S}_{\eta,m,n}(\boldsymbol{\beta}^{(1)},\boldsymbol{\beta}^{(2)})]\right|}_{\mathcal{I}_2} \\
=&S(\boldsymbol{\beta}^{(1)},\boldsymbol{\beta}^{(2)})\mathcal{O}\left(\frac{1}{n^2}\right) + S(\boldsymbol{\beta}^{(1)},\boldsymbol{\beta}^{(2)})\mathcal{O}\left(\frac{1}{n^2}\right),
\end{aligned} \tag{27}$$

where $\mathcal{I}_1$ and $\mathcal{I}_2$ follow from the proof of S1 without and with normality assumptions, respectively (Quiroz et al., 2019).

Combining (26) and (27), we have

$$\mathbb{E}[\min\{1, \widetilde{S}_{\eta,m,n}(\boldsymbol{\beta}^{(1)},\boldsymbol{\beta}^{(2)})\}] \approx \min\left\{1, S(\boldsymbol{\beta}^{(1)},\boldsymbol{\beta}^{(2)})\left(\mathcal{O}\left(\frac{1}{n^2}\right) + e^{-\mathcal{O}\left(\frac{m^2\eta}{n} + \frac{1}{n^2}\right)}\right)\right\}. \tag{28}$$

This means that reducing the update period $m$ (more frequent update the of control variable), the learning rate $\eta$ and the batch size $n$ significantly increases $\min\{1, \widetilde{S}_{\eta,m,n}\}$ on average. ∎

The above lemma shows a potential to exponentially increase the number of effective swaps via variance reduction under the same intensity $r$. Next, we show the impact of variance reduction in speeding up the exponential convergence of the corresponding continuous-time replica exchange Langevin diffusion.

**Theorem 1 (Exponential convergence)** *Under the smoothness and dissipativity assumptions 1 and 2, the replica exchange Langevin diffusion associated with the variance-reduced stochastic swapping rates $S_{\eta,m,n}(\cdot,\cdot) = \min\{1, \widetilde{S}_{\eta,m,n}(\cdot,\cdot)\}$ converges exponential fast to the invariant distribution $\pi$ given a smaller learning rate $\eta$, a smaller $m$ or a larger batch size $n$:*

$$\mathcal{W}_2(\nu_t, \pi) \leq D_0 \exp\left\{-t\left(1 + \delta_{S_{\eta,m,n}}\right)/c_{LS}\right\}, \tag{29}$$

*where $D_0 = \sqrt{2c_{LS}D(\nu_0||\pi)}$, $\delta_{S_{\eta,m,n}} := \inf_{t>0} \frac{\mathcal{E}_{S_{\eta,m,n}}(\sqrt{\frac{d\nu_t}{d\pi}})}{\mathcal{E}(\sqrt{\frac{d\nu_t}{d\pi}})} - 1$ is a non-negative constant depending on the truncated stochastic swapping rate $S_{\eta,m,n}(\cdot,\cdot)$ and increases with a smaller learning rate $\eta$, a shorter period $m$ and a large batch size $n$. $c_{LS}$ is the standard constant of the log-Sobolev inequality asscoiated with the Dirichlet form for replica exchange Langevin diffusion without swaps.*

**Proof** Given a smooth function $f : \mathbb{R}^d \times \mathbb{R}^d \to \mathbb{R}$, the infinitesimal generator $\mathcal{L}_{S_{\eta,m,n}}$ associated with the replica exchange Langevin diffusion with the swapping rate $S_{\eta,m,n} = \min\{1, \widetilde{S}_{\eta,m,n}\}$ follows

$$\begin{aligned}
\mathcal{L}_{S_{\eta,m,n}}f(\boldsymbol{\beta}^{(1)},\boldsymbol{\beta}^{(2)}) = &-\langle\nabla_{\boldsymbol{\beta}^{(1)}}f(\boldsymbol{\beta}^{(1)},\boldsymbol{\beta}^{(2)}), \nabla L(\boldsymbol{\beta}^{(1)})\rangle - \langle\nabla_{\boldsymbol{\beta}^{(2)}}f(\boldsymbol{\beta}^{(1)},\boldsymbol{\beta}^{(2)}), \nabla L(\boldsymbol{\beta}^{(2)})\rangle \\
&+ \tau^{(1)}\Delta_{\boldsymbol{\beta}^{(1)}}f(\boldsymbol{\beta}^{(1)},\boldsymbol{\beta}^{(2)}) + \tau^{(2)}\Delta_{\boldsymbol{\beta}^{(2)}}f(\boldsymbol{\beta}^{(1)},\boldsymbol{\beta}^{(2)}) \\
&+ rS_{\eta,m,n}(\boldsymbol{\beta}^{(1)},\boldsymbol{\beta}^{(2)}) \cdot (f(\boldsymbol{\beta}^{(2)},\boldsymbol{\beta}^{(1)}) - f(\boldsymbol{\beta}^{(1)},\boldsymbol{\beta}^{(2)})),
\end{aligned} \tag{30}$$

where $\nabla_{\boldsymbol{\beta}^{(h)}}$ and $\Delta_{\boldsymbol{\beta}^{(h)}}$ are the gradient and the Laplace operators with respect to $\boldsymbol{\beta}^{(h)}$, respectively. Next, we model the exponential decay of $\mathcal{W}_2(\nu_t, \pi)$ using the Dirichlet form

$$\mathcal{E}_{S_{\eta,m,n}}(f) = \int \Gamma_{S_{\eta,m,n}}(f)d\pi, \tag{31}$$

where $\Gamma_{S_{\eta,m,n}}(f) = \frac{1}{2} \cdot \mathcal{L}_{S_{\eta,m,n}}(f^2) - f\mathcal{L}_{S_{\eta,m,n}}(f)$ is the Carré du Champ operator. In particular for the first term $\frac{1}{2}\mathcal{L}_{S_{\eta,m,n}}(f^2)$, we have

$$
\begin{aligned}
&\frac{1}{2}\mathcal{L}_{S_{\eta,m,n}}(f(\boldsymbol{\beta}^{(1)}, \boldsymbol{\beta}^{(2)})^2) \\
=& -\langle f(\boldsymbol{\beta}^{(1)}, \boldsymbol{\beta}^{(2)})\nabla_{\boldsymbol{\beta}^{(1)}}f(\boldsymbol{\beta}^{(1)}, \boldsymbol{\beta}^{(2)}), \nabla_{\boldsymbol{\beta}^{(1)}}L(\boldsymbol{\beta}^{(1)})\rangle + \tau^{(1)}\|\nabla_{\boldsymbol{\beta}^{(1)}}f(\boldsymbol{\beta}^{(1)}, \boldsymbol{\beta}^{(2)})\|^2 \\
&\qquad\qquad\qquad\qquad\qquad + \tau^{(1)}f(\boldsymbol{\beta}^{(1)}, \boldsymbol{\beta}^{(2)})\Delta_{\boldsymbol{\beta}^{(1)}}f(\boldsymbol{\beta}^{(1)}, \boldsymbol{\beta}^{(2)}) \\
&- \langle f(\boldsymbol{\beta}^{(1)}, \boldsymbol{\beta}^{(2)})\nabla_{\boldsymbol{\beta}^{(2)}}f(\boldsymbol{\beta}^{(1)}, \boldsymbol{\beta}^{(2)}), \nabla_{\boldsymbol{\beta}^{(2)}}L(\boldsymbol{\beta}^{(2)})\rangle + \tau^{(2)}\|\nabla_{\boldsymbol{\beta}^{(2)}}f(\boldsymbol{\beta}^{(1)}, \boldsymbol{\beta}^{(2)})\|^2 \\
&\qquad\qquad\qquad\qquad\qquad + \tau^{(2)}f(\boldsymbol{\beta}^{(1)}, \boldsymbol{\beta}^{(2)})\Delta_{\boldsymbol{\beta}^{(2)}}f(\boldsymbol{\beta}^{(1)}, \boldsymbol{\beta}^{(2)}) \\
&+ \frac{r}{2}S_{\eta,m,n}(\boldsymbol{\beta}^{(1)}, \boldsymbol{\beta}^{(2)})(f^2(\boldsymbol{\beta}^{(2)}, \boldsymbol{\beta}^{(1)}) - f^2(\boldsymbol{\beta}^{(1)}, \boldsymbol{\beta}^{(2)})).
\end{aligned}
$$

Combining the definition of the Carré du Champ operator, (30) and (B), we have

$$
\begin{aligned}
&\Gamma_{S_{\eta,m,n}}(f(\boldsymbol{\beta}^{(1)}, \boldsymbol{\beta}^{(2)})) \\
=&\frac{1}{2}\mathcal{L}_{S_{\eta,m,n}}(f^2(\boldsymbol{\beta}^{(1)}, \boldsymbol{\beta}^{(2)})) - f(\boldsymbol{\beta}^{(1)}, \boldsymbol{\beta}^{(2)})\mathcal{L}_{S_{\eta,m,n}}(f(\boldsymbol{\beta}^{(1)}, \boldsymbol{\beta}^{(2)})) \\
=&\tau^{(1)}\|\nabla_{\boldsymbol{\beta}^{(1)}}f(\boldsymbol{\beta}^{(1)}, \boldsymbol{\beta}^{(2)})\|^2 + \tau^{(2)}\|\nabla_{\boldsymbol{\beta}^{(2)}}f(\boldsymbol{\beta}^{(1)}, \boldsymbol{\beta}^{(2)})\|^2 \\
&+ \frac{r}{2}S_{\eta,m,n}(\boldsymbol{\beta}^{(1)}, \boldsymbol{\beta}^{(2)})(f(\boldsymbol{\beta}^{(2)}, \boldsymbol{\beta}^{(1)}) - f(\boldsymbol{\beta}^{(1)}, \boldsymbol{\beta}^{(2)}))^2.
\end{aligned}
\tag{32}
$$

Plugging (32) into (31), the Dirichlet form associated with operator $\mathcal{L}_{S_{\eta,m,n}}$ follows

$$
\begin{aligned}
\mathcal{E}_{S_{\eta,m,n}}(f) = &\underbrace{\int \left(\tau^{(1)}\|\nabla_{\boldsymbol{\beta}^{(1)}}f(\boldsymbol{\beta}^{(1)}, \boldsymbol{\beta}^{(2)})\|^2 + \tau^{(2)}\|\nabla_{\boldsymbol{\beta}^{(2)}}f(\boldsymbol{\beta}^{(1)}, \boldsymbol{\beta}^{(2)})\|^2\right)d\pi(\boldsymbol{\beta}^{(1)}, \boldsymbol{\beta}^{(2)})}_{\text{vanilla term } \mathcal{E}(f)} \\
&+ \underbrace{\frac{r}{2}\int S_{\eta,m,n}(\boldsymbol{\beta}^{(1)}, \boldsymbol{\beta}^{(2)}) \cdot (f(\boldsymbol{\beta}^{(2)}, \boldsymbol{\beta}^{(1)}) - f(\boldsymbol{\beta}^{(1)}, \boldsymbol{\beta}^{(2)}))^2 d\pi(\boldsymbol{\beta}^{(1)}, \boldsymbol{\beta}^{(2)})}_{\text{acceleration term}},
\end{aligned}
\tag{33}
$$

where $f$ corresponds to $\frac{d\nu_t}{d\pi(\boldsymbol{\beta}^{(1)}, \boldsymbol{\beta}^{(2)})}$. Under the asymmetry conditions of $\frac{\nu_t}{\pi(\boldsymbol{\beta}_1, \boldsymbol{\beta}^{(2)})}$ and $S_{\eta,m,n} > 0$, the acceleration term of the Dirichlet form is strictly positive and linearly dependent on the swapping rate $S_{\eta,m,n}$. Therefore, $\mathcal{E}_{S_{\eta,m,n}}(f)$ becomes significantly larger as the swapping rate $S_{\eta,m,n}$ increases significantly. According to Lemma 5 (Deng et al., 2020), there exists a constant $\delta_{S_{\eta,m,n}} = \inf_{t>0}\frac{\mathcal{E}_{S_{\eta,m,n}}(\sqrt{\frac{d\nu_t}{d\pi}})}{\mathcal{E}(\sqrt{\frac{d\nu_t}{d\pi}})} - 1$ depending on $S_{\eta,m,n}$ that satisfies the following log-Sobolev inequality for the unique invariant measure $\pi$ associated with variance-reduced replica exchange Langevin diffusion $\{\boldsymbol{\beta}_t\}_{t\geq 0}$

$$D(\nu_t||\pi) \leq 2\frac{c_{\text{LS}}}{1 + \delta_{S_{\eta,m,n}}}\mathcal{E}_{S_{\eta,m,n}}\left(\sqrt{\frac{d\nu_t}{d\pi}}\right),$$

where $\delta_{S_{\eta,m,n}}$ increases rapidly with the swapping rate $S_{\eta,m,n}$. By virtue of the exponential decay of entropy (Bakry et al., 2014), we have

$$D(\nu_t||\pi) \leq D(\nu_0||\pi)e^{-2t(1+\delta_{S_{\eta,m,n}})/c_{\text{LS}}},$$

where $c_{\text{LS}}$ is the standard constant of the log-Sobolev inequality asscoiated with the Dirichlet form for replica exchange Langevin diffusion without swaps (Lemma 4 as in Deng et al. (2020)). Next, we upper bound $\mathcal{W}_2(\nu_t, \pi)$ by the Otto-Villani theorem (Bakry et al., 2014)

$$\mathcal{W}_2(\nu_t, \pi) \leq \sqrt{2c_{\text{LS}}D(\nu_t||\pi)} \leq \sqrt{2c_{\text{LS}}D(\mu_0||\pi)}e^{-t(1+\delta_{S_{\eta,m,n}})/c_{\text{LS}}},$$

where $\delta_{S_{\eta,m,n}} > 0$ depends on the learning rate $\eta$, the period $m$ and the batch size $n$. ∎

In the above analysis, we have established the relation that $\delta_{S_{\eta,m,n}} = \inf_{t>0} \frac{\mathcal{E}_{S_{\eta,m,n}}(\sqrt{\frac{d\nu_t}{d\pi}})}{\mathcal{E}(\sqrt{\frac{d\nu_t}{d\pi}})} - 1$

depending on $S_{\eta,m,n}$ may increase significantly with a smaller learning rate $\eta$, a shorter period $m$ and a large batch size $n$. For more quantitative study on how large $\delta_{S_{\eta,m,n}}$ is on related problems, we refer interested readers to the study of spectral gaps in Lee et al. (2018); Dong & Tong (2020); Futami et al. (2020).

## C  DISCRETIZATION ERROR

Consider a complete filtered probability space $(\Omega, \mathcal{F}, \mathbb{F} = (\mathcal{F}_t)_{t\in[0,T]}, \mathbb{P})$ which supports all the random subjects considered in the sequel. With a little abuse usage of notation, the probability measure $\mathbb{P}$ (component wise if $\mathbb{P}$ is joint probability measure with mutually independent components) would always denote the Wiener measure under which the process $(\boldsymbol{W}_t)_{0\leq t\leq T}$ is a $\mathbb{P}$-Brownian motion. To be precise, in what follows, we shall denote $\mathbb{P} := \mathbb{P}^{\boldsymbol{W}} \times \mathbf{N}$, where $\mathbb{P}^{\boldsymbol{W}}$ is the infinite dimensional Wiener measure and $\mathbf{N}$ is the Poisson measure independent of $\mathbb{P}^{\boldsymbol{W}}$ and has some constant jump intensity. In our general framework below, the jump process $\alpha$ is introduced by swapping the diffusion matrix of the two Langevin dynamics and the jump intensity is defined through the swapping probability in the following sense, which ensures the independence of $\mathbb{P}^{\boldsymbol{W}}$ and $\mathbf{N}^S$ in each time interval $[i\eta, (i+1)\eta]$, for $i \in \mathbb{N}^+$. The precise definition of the **Replica exchange Langevin diffusion (reLD)** is given as below. For any fixed learning rate $\eta > 0$, we define

$$\begin{cases} d\boldsymbol{\beta}_t = -\nabla G(\boldsymbol{\beta}_t)dt + \Sigma(\alpha_t)d\boldsymbol{W}_t, \\ \mathbb{P}\left(\alpha(t) = j | \alpha(t-dt) = l, \boldsymbol{\beta}(\lfloor t/\eta\rfloor\eta) = \boldsymbol{\beta}\right) = rS(\boldsymbol{\beta})\eta\mathbf{1}_{\{t=\lfloor t/\eta\rfloor\eta\}} + o(dt), \text{ for } l \neq j, \end{cases} \tag{34}$$

where $\nabla G(\boldsymbol{\beta}) := \begin{pmatrix} \nabla L(\boldsymbol{\beta}^{(1)}) \\ \nabla L(\boldsymbol{\beta}^{(2)}) \end{pmatrix}$, and $\mathbf{1}_{t=\lfloor t/\eta\rfloor\eta}$ is the indicator function, i.e. for every $t = i\eta$ with $i \in \mathbb{N}^+$, given $\boldsymbol{\beta}(i\eta) = \boldsymbol{\beta}$, we have $\mathbb{P}\left(\alpha(t) = j | \alpha(t-dt) = l\right) = rS(\boldsymbol{\beta})\eta$, where $S(\boldsymbol{\beta})$ is defined as $\min\{1, S(\boldsymbol{\beta}^{(1)}, \boldsymbol{\beta}^{(2)})\}$ and $S(\boldsymbol{\beta}^{(1)}, \boldsymbol{\beta}^{(2)})$ is defined in (13). In this case, the Markov Chain $\alpha(t)$ is a constant on the time interval $[\lfloor t/\eta\rfloor\eta, \lfloor t/\eta\rfloor\eta + \eta)$ with some state in the finite-state space $\{0, 1\}$ and the generator matrix $Q$ follows

$$Q = \begin{pmatrix} -rS(\boldsymbol{\beta})\eta\delta(t - \lfloor t/\eta\rfloor\eta) & rS(\boldsymbol{\beta})\eta\delta(t - \lfloor t/\eta\rfloor\eta) \\ rS(\boldsymbol{\beta})\eta\delta(t - \lfloor t/\eta\rfloor\eta) & -rS(\boldsymbol{\beta})\eta\delta(t - \lfloor t/\eta\rfloor\eta) \end{pmatrix},$$

where $\delta(\cdot)$ is a Dirac delta function. The diffusion matrix $\Sigma(\alpha_t)$ is thus defined as $(\Sigma(0), \Sigma(1)) :=$ $\left\{ \begin{pmatrix} \sqrt{2\tau^{(1)}}\mathbf{I}_d & 0 \\ 0 & \sqrt{2\tau^{(2)}}\mathbf{I}_d \end{pmatrix}, \begin{pmatrix} \sqrt{2\tau^{(2)}}\mathbf{I}_d & 0 \\ 0 & \sqrt{2\tau^{(1)}}\mathbf{I}_d \end{pmatrix} \right\}$. From our definition and following Yin & Zhu (2010)[Section 2.7], the generator matrix $Q$ will depend on the initial value at each time interval $[i\eta, (i+1)\eta]$. The distribution of process $(\boldsymbol{\beta}_t)_{0\leq t\leq T}$ is denoted as $\nu_T := \mathbb{P}^G \times \mathbf{N}^S$ which is absolutely continuous with respect to the reference measure $\mathbb{P} := \mathbb{P}^{\boldsymbol{W}} \times \mathbf{N}$, under which $\boldsymbol{W}$ is Brownian motion and $\alpha(\cdot)$ is a Poisson process with some constant jump intensity. This fact follows from the result in Gikhman & Skorokhod (1980)[VII, Section 6, Theorem 2] and Yin & Zhu (2010)[Section 2.5, formula (2.40)]. The motivation of only considering the positive swapping rate in $i\eta$, for $i \in \mathbb{N}^+$, and zero elsewhere is due to our construction of the discretized process $\widetilde{\boldsymbol{\beta}}$ as shown below (see equation 35). A simple illustration of the idea can be seen from the auxiliary process construction in Yin & Zhu (2010)[Section 2.5], following which we want to make sure the stopping time of $\boldsymbol{\beta}$ and $\widetilde{\boldsymbol{\beta}}$ happening at the same time. Otherwise, it is unlikely (and also unreasonable) to derive the Radon-Nikodym derivative of the two process $\boldsymbol{\beta}$ and $\widetilde{\boldsymbol{\beta}}$. Thus, we should think of the process is concatenated on the time interval $[i\eta, (i+1)\eta]$ up to time horizon $T$. Similarly, we consider the following **Replica exchange stochastic gradient Langevin diffusion**, for the same learning rate $\eta > 0$ as above, we have

$$\begin{cases} d\widetilde{\boldsymbol{\beta}}_t^\eta = -\nabla\widetilde{G}(\widetilde{\boldsymbol{\beta}}_{\lfloor t/\eta\rfloor\eta}^\eta)dt + \Sigma(\widetilde{\alpha}_{\lfloor t/\eta\rfloor\eta})d\boldsymbol{W}_t, \\ \mathbb{P}\left(\widetilde{\alpha}(t) = j | \widetilde{\alpha}(t-dt) = l, \widetilde{\boldsymbol{\beta}}(\lfloor t/\eta\rfloor\eta) = \widetilde{\boldsymbol{\beta}}\right) = r\widetilde{S}(\widetilde{\boldsymbol{\beta}})\eta\mathbf{1}_{\{t=\lfloor t/\eta\rfloor\eta\}} + o(dt), \text{ for } l \neq j, \end{cases} \tag{35}$$

where $\nabla\widetilde{G}(\boldsymbol{\beta}) := \begin{pmatrix} \nabla\widetilde{L}(\boldsymbol{\beta}^{(1)}) \\ \nabla\widetilde{L}(\boldsymbol{\beta}^{(2)}) \end{pmatrix}$ and $\widetilde{S}(\widetilde{\boldsymbol{\beta}}) = \min\{1, \widetilde{S}_{\eta,m,n}(\widetilde{\boldsymbol{\beta}}^{(1)}, \widetilde{\boldsymbol{\beta}}^{(2)})\}$ and $\widetilde{S}_{\eta,m,n}(\widetilde{\boldsymbol{\beta}}^{(1)}, \widetilde{\boldsymbol{\beta}}^{(2)})$ is

shown in (16). The distribution of process $(\widetilde{\boldsymbol{\beta}}_t)_{0 \leq t \leq T}$ is denoted as $\mu_T := \mathbb{P}^{\widetilde{G}} \times \mathbf{N}^{\widetilde{S}}$, where $\widetilde{\alpha}$ is a Poisson process with jump intensity $r\widetilde{S}(\widetilde{\boldsymbol{\beta}})\eta\delta(t - \lfloor t/\eta \rfloor \eta)$ on the time interval $[\lfloor t/\eta \rfloor \eta, \lfloor t/\eta \rfloor \eta + \eta)$. Note that $\boldsymbol{\beta}$ and $\widetilde{\boldsymbol{\beta}}$ are defined by using the same $\mathbb{P}$-Brownian motion $\boldsymbol{W}$, but with two different jump intensity on the time interval $[\lfloor t/\eta \rfloor \eta, \lfloor t/\eta \rfloor \eta + \eta)$. Notice that, if there is no jump, the construction of $\widetilde{\boldsymbol{\beta}}$ based on $\boldsymbol{\beta}$ follows from the fact that they share the same marginal distributions as shown in Gyöngy (1986), where one can find the details in Raginsky et al. (2017). Given the jump process $\alpha$ and $\widetilde{\alpha}$ introduced into the dynamics of $\boldsymbol{\beta}$ and $\widetilde{\boldsymbol{\beta}}$, the construction is more complicated. Thanks to Bentata & Cont (2009), we can carry on the similar construction in our current setting. We then introduce the following Radon-Nikodym density for $d\nu_T/d\mu_T$. In the current setting, the change of measure can be seen as the combination of two drift-diffusion process and two jump process simultaneously. We first introduce some notation. For each vector $A \in \mathbb{R}^n$, we denote $\|A\|^2 := A^*A$. Furthermore, we introduce a sequence of stopping time based on our definition of process $\boldsymbol{\beta}$ and $\widetilde{\boldsymbol{\beta}}$. For $j \in \mathbb{N}^+$, we denote $\zeta_j's$ as a stopping times defined by $\zeta_{j+1} := \inf\{t > \zeta_j : \alpha(t) \neq \alpha(\zeta_j)\}$ and $N(T) = \max\{n \in \mathbb{N} : \zeta_n \leq T\}$. It is easy to see that for any stopping time $\zeta_i$, there exists $l \in \mathbb{N}^+$ such that $\zeta_j = l\eta$. Similarly, we have the stopping time for the process $\widetilde{\boldsymbol{\beta}}$ by $\widetilde{\zeta}_{j+1} := \inf\{t > \widetilde{\zeta}_j : \widetilde{\alpha}(t) \neq \widetilde{\alpha}(\zeta_j)\}$ and $\widetilde{\alpha}(t)$ follows the same trajectory of $\alpha(t)$. To serve the purpose of our analysis, one should think of the process $\boldsymbol{\beta}$ as the auxiliary process to the process $\widetilde{\boldsymbol{\beta}}$, see similar constructions in Yin & Zhu (2010)[Section 2.5, formula (2.39)]. The difference is that both of our process $\boldsymbol{\beta}$ and $\widetilde{\boldsymbol{\beta}}$ are associated with jump process jumping at time $i\eta$, for some integer $i \in \mathbb{N}^+$, instead of jumping at any continuous time. We combine approximation method from Yin & Zhu (2010)[Section 2.7] for non-constant generator matrix $Q$ and the density representation for Markov process in Gikhman & Skorokhod (1980)[VII, Section 6, Teorem 2] to get the following

**Lemma C1** *Let $\{\zeta_j | j \in \{0, 1, \cdots, N(T)\}\}$ be a sequence of stopping time defined by $\alpha$. Let $k \in \mathbb{N}^+$ be an fixed integer such that $k\eta \leq T \leq (k+1)\eta$. For each fixed learning rate $\eta > 0$ and for any $\varepsilon > 0$, the Radon-Nikodym derivative of $d\mu_T/d\nu_T$ is given below,*

$$\frac{d\mu_T}{d\nu_T} = \exp\Big(\sum_{j=0}^{N(T)}\int_{\zeta_j}^{\zeta_{j+1}\wedge T}\Big[\Sigma^{-1}(\widetilde{\alpha}(\zeta_j))\nabla\widetilde{G}(\boldsymbol{\beta}_t) - \Sigma^{-1}(\alpha(\zeta_j))\nabla G(\boldsymbol{\beta}_t)\Big]d\boldsymbol{W}_t^G$$

$$-\frac{1}{2}\sum_{j=0}^{N(T)}\int_{\zeta_j}^{\zeta_{j+1}\wedge T}\Big\|\Sigma^{-1}(\widetilde{\alpha}(\zeta_j))\nabla\widetilde{G}(\boldsymbol{\beta}_t) - \Sigma^{-1}(\alpha(\zeta_j))\nabla G(\boldsymbol{\beta}_t)\Big\|^2 dt\Big)$$

$$\times \exp\Big\{-\sum_{j=0}^{N(T))}\int_{\zeta_j}^{\zeta_{j+1}\wedge T-\varepsilon} r\delta(t - \lfloor t/\eta \rfloor \eta)[\widetilde{S}(\widetilde{\boldsymbol{\beta}}_{\lfloor t/\eta \rfloor \eta}) - S(\boldsymbol{\beta}_{\lfloor t/\eta \rfloor \eta})]\eta dt\Big\} \times \Pi_{j=0}^{N(T)}\frac{\widetilde{S}(\widetilde{\boldsymbol{\beta}}_{\zeta_j})}{S(\boldsymbol{\beta}_{\zeta_j})}.$$

**Proof** Recall that $\zeta_j$ is stopping time defined by $\alpha$ (same as defined by $\widetilde{\alpha}$), i.e. $\zeta_{j+1} := \inf\{t > \zeta_j : \alpha(t) \neq \alpha(\zeta_j)\}$, for $j = 0, 1, \cdots, N(T)$, and for each $\zeta_j$, there exists $l \in \{0, 1, \cdots, k\}$ such that $\zeta_j = l\eta$. We now follow Gikhman & Skorokhod (1980)[VII, Section 6, Theorem 2] to derive the Radon-Nikodym density for $d\mu_T/d\nu_T$. In this case, if the generator matrix $Q$ is constant, i.e. the jump intensity is constant, we can follow the similar construction from Yin & Zhu (2010)[Formula (2.40)], see also Eizenberg & Freidlin (1990)[Formula(3.13)]. Next, we adjust our setting to the case that we can treat our generator matrix as constant on each time interval $[\zeta_j, \zeta_{j+1})$, then the existing results apply to our case for the density with respect to the Poisson measure (jump process $\alpha$ and $\widetilde{\alpha}$), i.e. $d\mathbf{N}^S/d\mathbf{N}^{\widetilde{S}}$. Furthermore, once the generator matrix $Q$ is constant, then the measure $\mathbb{P}^G$ ( or $\mathbb{P}^{\widetilde{G}}$) is independent to $\mathbf{N}^S$ (or $\mathbf{N}^{\widetilde{S}}$). We show the following steps to give a clear outline of our proof.

**Step 1:** For each stopping time interval $[\zeta_j, \zeta_{j+1})$, no jump would occur after the initial point at time $\zeta_j$ and the diffusion matrix $\Sigma$ and $\widetilde{\Sigma}$ keep the same, thus we can apply the generalized Girsanov theorem to get the Randon-Nikodym derivative for $d\mathbb{P}^G/d\mathbb{P}^{\widetilde{G}}$.

**Step 2:** In order to combine the the two density of $d\mathbf{N}^S/d\mathbf{N}^{\widetilde{S}}$ and $d\mathbb{P}^G/d\mathbb{P}^{\widetilde{G}}$, we need the independent property of the two measures on the same time interval, then we directly get the density

following from Gikhman & Skorokhod (1980)[VII, Section 6, Theorem 2]. Different from the work mentioned above, we will first write all the density on each time interval $[i\eta, (i + 1)\eta)$ to incorporate the independent requirement mentioned above. Notice that the relative change of density for $d\mathbf{N}^S/d\mathbf{N}^{\widetilde{S}}$ would only depends on the left end point, since the jump intensity would change its values at the initial value of interval $[i\eta, (i + 1)\eta)$, which is a standard idea to deal with generator matrix $Q$ depending on the initial value instead of a constant matrix case. (See Yin & Zhu (2010)[Section2.7] for similar treatments).

**Step 3:** In general, the stopping time interval could contain several time interval with length $\eta$, however the jump intensity should only depend on the left end point for each time interval $[i\eta, (i + 1)\eta)$. Based on the above set up, we now derive the Radon-Nikodym derivative. First notice that, on each period $[\zeta_j, \zeta_{j+1})$, the matrix $\Sigma$ is fixed and is evaluated at $\Sigma(\alpha(\zeta_j))$, which is the same for $\Sigma(\widetilde{\alpha}(\zeta_j))$. In particular, $\Sigma(\alpha(\zeta_j)) = \Sigma(\widetilde{\alpha}(\zeta_j))$ is a constant diagonal matrix. According to our definition $d\nu_T = d\mathbb{P}^G \times d\mathbf{N}^S$ and $d\mu_T = d\mathbb{P}^{\widetilde{G}} \times d\mathbf{N}^{\widetilde{S}}$, we write the Radon-Nikodym derivative on each of the time interval $[i\eta, (i + 1)\eta)$ and concatenate them together. We consider the swapping of the diffusion matrix first where a similar construction can be found in Yin & Zhu (2010)[Formula (2.40)], we get the following Radon-Nikodym derivative, for any $\varepsilon > 0$,

$$\frac{d\mathbf{N}^{\widetilde{S}}}{d\mathbf{N}^S} = \exp\Big\{ -\sum_{j=0}^{N(T)} \int_{j\eta}^{(j+1)\eta \wedge T - \varepsilon} r\delta(t - \lfloor t/\eta \rfloor \eta)(\widetilde{S}(\boldsymbol{\beta}_{\boldsymbol{\beta}_{\lfloor t/\eta \rfloor \eta}}) - S(\boldsymbol{\beta}_{\boldsymbol{\beta}_{\lfloor t/\eta \rfloor \eta}}))\eta dt \Big\} \quad (36)$$
$$\times \Pi_{j=0}^{N(T)} \frac{\widetilde{S}(\widetilde{\boldsymbol{\beta}}_{\zeta_j})}{S(\boldsymbol{\beta}_{\zeta_j})}.$$

Next, we show the density for $d\mathbb{P}^G/d\mathbb{P}^{\widetilde{G}}$ as below. On each interval $[\zeta_j, \zeta_{j+1})$, given initial value $(\boldsymbol{\beta}_j, \widetilde{\boldsymbol{\beta}}_j)$, the matrix $\Sigma(\alpha(\zeta_j))$ and $\Sigma(\widetilde{\alpha}(\zeta_j))$ are always the same, since no jump would happen. In particular, in this continuous case the integral on $[\zeta_j, \zeta_{j+1})$ and $[\zeta_j, \zeta_{j+1}]$ are the same. Thus we have the following Radon-Nikodym derivative

$$\frac{d\mathbb{P}^{\widetilde{G}}}{d\mathbb{P}^G} = \exp\Big( \sum_{j=0}^{N(T)} \int_{\zeta_j}^{\zeta_{j+1} \wedge T} \Big[ \Sigma^{-1}(\widetilde{\alpha}(\zeta_j))\nabla\widetilde{G}(\boldsymbol{\beta}_t) - \Sigma^{-1}(\alpha(\zeta_j))\nabla G(\boldsymbol{\beta}_t) \Big] d\boldsymbol{W}_t^G$$
$$-\frac{1}{2} \sum_{j=0}^{N(T)} \int_{\zeta_j}^{\zeta_{j+1} \wedge T} \Big\| \Sigma^{-1}(\widetilde{\alpha}(\zeta_j))\nabla\widetilde{G}(\boldsymbol{\beta}_t) - \Sigma^{-1}(\alpha(\zeta_j))\nabla G(\boldsymbol{\beta}_t) \Big\|^2 dt \Big). \quad (37)$$

Notice that matrix $\Sigma$ is diagonal square matrix, thus we have $\Sigma = \Sigma^*$. Recall that $\boldsymbol{W}$ is a $\mathbb{P}$-Brownian motion, assuming there is no jump in the dynamic for $\boldsymbol{\beta}$, then according to the Girsanov theorem (see an example in Theorem 8.6.6 and Example 8.6.9 (Øksendal, 2003)) with Radon-Nikodym derivative $d\mathbb{P}^G/d\mathbb{P}$, we have the $\mathbb{P}^G$-Brownian motion, denoted as $\boldsymbol{W}^G$, which follows

$$\boldsymbol{W}_t^G := \boldsymbol{W}_t + \int_0^t \Sigma^{-1}(\alpha_s)(\nabla G(\boldsymbol{\beta}_s))ds. \quad (38)$$

This fact holds true on each of the time interval $[\zeta_j, \zeta_{j+1}]$. Multiplying the two density $d\mathbb{P}^G/\mathbb{P}^{\widetilde{G}}$ and $d\mathbf{N}^S/d\mathbf{N}^{\widetilde{S}}$, we complete the proof.

**Remark 1** *Notice that, if we keep the constant diffusion matrix without jump, then the Randon-Nikodym derivative $d\mu_T/d\nu_T$ has been used in the stochastic gradient descent setting, for example Raginsky et al. (2017). However, the notation of the Brownian motion has been used freely, we try to make it consistent in the current setting. Namely, for constant diffusion matrix $\Sigma$, we have*

$$\frac{d\mathbb{P}^{\widetilde{G}}}{d\mathbb{P}^G} = \exp\Big( \int_0^T \Big[ \Sigma^{-1}\nabla\widetilde{G}(\widetilde{\boldsymbol{\beta}}_s) - \Sigma^{-1}\nabla G(\boldsymbol{\beta}_s) \Big] d\boldsymbol{W}_s^G$$
$$-\frac{1}{2} \int_0^T \Big\| \Sigma^{-1}\nabla\widetilde{G}(\widetilde{\boldsymbol{\beta}}_s) - \Sigma^{-1}\nabla G(\boldsymbol{\beta}_s) \Big\|^2 ds \Big), \quad (39)$$

*where $\boldsymbol{W}^G$ is a $\mathbb{P}^G$-Brownian motion as shown in equation 38, not a $\mathbb{P}$-Brownian motion $\boldsymbol{W}$.*

**Remark 2** *The density $\frac{d\mu_T}{d\nu_T}$ that we derived above is so far the best we can do. If one would like to use the continuous time control $\alpha(t)$ with continuous jump intensity $S(\beta(t))$ instead of jumping at the initial point with a fixed rate, then we can not even write the Randon-Nikodym derivative anymore, since $\alpha(t)$ and $\widetilde{\alpha}(t)$ will define different stopping time, i.e. jump at different time and $\mu_T$ is not absolutely continuous with respect to $\nu_T$.*

Based on the above lemma, we further get the following estimates.

**Lemma C2** *Given a large enough batch size $n$ or a small enough $m$ and $\eta$, we have the bound of the KL divergence of $D_{KL}(\mu_T|\nu_T)$ as below,*

$$D_{KL}(\mu_T|\nu_T) \leq (\Phi_0 + \Phi_1\eta)k\eta + N(T)\Phi_2,$$

*with*

$$\Phi_0 = \mathcal{O}\left(\frac{m}{\sqrt{n}}\sqrt{\eta}d\right) + \frac{r\delta\Phi^2}{4\tau^{(1)}},$$

$$\Phi_1 = \left(C^2d\frac{\tau^{(2)}}{\tau^{(1)}} + \frac{C^2\delta kd}{2\tau^{(1)}}[\tau^{(1)} + \tau^{(2)}]\right),$$

$$\Phi_2 = \mathcal{O}\left(\frac{m}{\sqrt{n}}\sqrt{\eta}d\right).$$

**Proof** By the very definition of the KL-divergence, we have

$$D_{KL}(\mu_T|\nu_T) = -\int d\nu_T \log\frac{d\mu_T}{d\nu_T}$$

$$= -\mathbb{E}_{\nu_T}\left[\log(d\mu_T/d\nu_T)\Big|(\boldsymbol{\beta}, \widetilde{\boldsymbol{\beta}}) = (\beta, \widetilde{\beta})\right].$$

We shall keep the convention below and denote $\mathbb{E}_{\nu_T,\beta} = \mathbb{E}_{\nu_T}[\cdot|(\boldsymbol{\beta}, \widetilde{\boldsymbol{\beta}}) = (\beta, \widetilde{\beta})]$, where $\beta = (\beta^{(1)}, \beta^{(2)}) \in \mathbb{R}^{2d}$ and $\widetilde{\beta} = (\widetilde{\beta}^{(1)}, \widetilde{\beta}^{(2)}) \in \mathbb{R}^{2d}$ denotes the values at each time $i\eta$, $i = 0, 1, \cdots, k$. Plugging Lemma C1 in the above equation and we unify the notation by using time intervals of the type $[i\eta, (i+1)\eta]$. To be precise, we get

$$\frac{d\mathbb{P}^{\widetilde{G}}}{d\mathbb{P}^G} = \exp\Big(\sum_{i=0}^{k-1}\int_{i\eta}^{(i+1)\eta}\left[\Sigma^{-1}(\widetilde{\alpha}(i\eta))\nabla\widetilde{G}(\boldsymbol{\beta}_t) - \Sigma^{-1}(\alpha(i\eta))\nabla G(\boldsymbol{\beta}_t)\right]d\boldsymbol{W}_t^G$$

$$+ \int_{k\eta}^{T}\left[\Sigma^{-1}(\widetilde{\alpha}(k\eta))\nabla\widetilde{G}(\boldsymbol{\beta}_t) - \Sigma^{-1}(\alpha(k\eta))\nabla G(\boldsymbol{\beta}_t)\right]d\boldsymbol{W}_t^G$$

$$- \frac{1}{2}\sum_{i=0}^{k-1}\int_{i\eta}^{(k+1)\eta}\left\|\Sigma^{-1}(\widetilde{\alpha}(i\eta))\nabla\widetilde{G}(\boldsymbol{\beta}_t) - \Sigma^{-1}(\alpha(i\eta))\nabla G(\boldsymbol{\beta}_t)\right\|^2 dt$$

$$- \frac{1}{2}\int_{k\eta}^{T}\left\|\Sigma^{-1}(\widetilde{\alpha}(k\eta))\nabla\widetilde{G}(\boldsymbol{\beta}_t) - \Sigma^{-1}(\alpha(k\eta))\nabla G(\boldsymbol{\beta}_t)\right\|^2 dt\Big). \tag{40}$$

The above equality follows from the fact that each time interval $[\zeta_j, \zeta_{j+1}]$ always contain exactly some sub-interval $[i\eta, (i+1)\eta]$. Namely, we have $[\zeta_j, \zeta_{j+1}] = [i\eta, (i+1)\eta] \cup [(j+1)\eta, (j+2)\eta] \cup \cdots \cup [l\eta, (l+1)\eta]$, for some $i, l \in \{0, 1, \cdots, k\}$. In particular, the matrix $\Sigma$ keep the same on each interval $[i\eta, (i+1)\eta]$, for some $i \in \{0, 1, \cdots, k\}$. Similarly, we expand the Radon-Nokodym derivative for $\frac{d\mathbf{N}^{\widetilde{S}}}{d\mathbf{N}^S}$ on the time interval of length $\eta$. Based on our definition of jump intensity, we get

$$\frac{d\mathbf{N}^{\widetilde{S}}}{d\mathbf{N}^S} = \exp\Big\{-\sum_{j=0}^{N(T)}\int_{j\eta}^{(j+1)\eta\wedge T-\varepsilon} r\delta(t - \lfloor t/\eta\rfloor\eta)(\widetilde{S}(\widetilde{\boldsymbol{\beta}}_{\lfloor t/\eta\rfloor\eta}) - S(\boldsymbol{\beta}_{\lfloor t/\eta\rfloor\eta}))\eta dt$$

$$- \int_{k\eta}^{T} r\delta(s - \lfloor s/\eta\rfloor\eta)(\widetilde{S}(\widetilde{\boldsymbol{\beta}}_{k\eta}) - S(\boldsymbol{\beta}_{k\eta}))\eta ds\Big\} \times \Pi_{j=0}^{N(T)}\left(\frac{\widetilde{S}(\widetilde{\boldsymbol{\beta}}_{\zeta_j})}{S(\boldsymbol{\beta}_{\zeta_j})}\right)$$

$$= \exp\Big\{-\sum_{i=0}^{k} r(\widetilde{S}(\widetilde{\boldsymbol{\beta}}_{i\eta}) - S(\boldsymbol{\beta}_{i\eta}))\eta\Big\} \times \Pi_{j=0}^{N(T)}\left(\frac{\widetilde{S}(\widetilde{\boldsymbol{\beta}}_{\zeta_j})}{S(\boldsymbol{\beta}_{\zeta_j})}\right). \tag{41}$$

Without loss of generality, we shall only consider the sum $\sum_{i=0}^{k-1}$ and skip the interval $[k\eta, T]$. Notice that on each time interval $[i\eta, (i+1)\eta)$, the control $\alpha(i\eta)$ and $\widetilde{\alpha}(i\eta)$ are fixed, thus the two component of the measure $d\nu_{T,\boldsymbol{\beta}}$ are independent. Taking into account the fact that $\boldsymbol{W}^G$ is $\mathbb{P}^G$-Brownian motion, thus we apply the martingale property and arrive at

$$
\begin{aligned}
& D_{KL}(\mu_T | \nu_T) \\
=& \mathbb{E}_{\nu_T, \boldsymbol{\beta}} \Big[ \frac{1}{2} \sum_{i=0}^{k-1} \int_{i\eta}^{(i+1)\eta} \Big\| \Sigma^{-1}(\widetilde{\alpha}(i\eta)) \nabla \widetilde{G}(\boldsymbol{\beta}_t) - \Sigma^{-1}(\alpha(i\eta)) \nabla G(\boldsymbol{\beta}_t) \Big\|^2 dt \Big] \\
& + \mathbb{E}_{\nu_T, \boldsymbol{\beta}} \Big[ \sum_{i=0}^{k-1} [\widetilde{S}(\widetilde{\boldsymbol{\beta}}_{i\eta}) - S(\boldsymbol{\beta}_{i\eta})]\eta - \sum_{j=0}^{N(T)} \Big( \log \widetilde{S}(\widetilde{\boldsymbol{\beta}}_{\zeta_j}) - \log S(\boldsymbol{\beta}_{\zeta_j}) \Big) \Big] \\
\leq& \underbrace{\frac{1}{2} \sum_{i=0}^{k-1} \mathbb{E}_{\nu_T, \boldsymbol{\beta}} \Big[ \int_{i\eta}^{(i+1)\eta} \Big\| \Sigma^{-1}(\widetilde{\alpha}(i\eta)) \nabla \widetilde{G}(\boldsymbol{\beta}_t) - \Sigma^{-1}(\alpha(i\eta)) \nabla G(\boldsymbol{\beta}_t) \Big\|^2 dt \Big]}_{\mathcal{I}} \\
& + \underbrace{\sum_{i=0}^{k-1} \mathbb{E}_{\nu_T, \boldsymbol{\beta}} \Big[ r | \widetilde{S}(\widetilde{\boldsymbol{\beta}}_{i\eta}) - S(\boldsymbol{\beta}_{i\eta}) | \eta \Big]}_{\mathcal{J}} + \underbrace{\sum_{j=0}^{N(T)} \mathbb{E}_{\nu_T, \boldsymbol{\beta}} \Big[ | \log \widetilde{S}(\widetilde{\boldsymbol{\beta}}_{\zeta_j}) - \log S(\boldsymbol{\beta}_{\zeta_j}) | \Big]}_{\mathcal{K}}.
\end{aligned}
$$

We then estimates the three terms $\mathcal{I}, \mathcal{J}, \mathcal{K}$ in order as below.

**Estimate of $\mathcal{I}$:** Due to the fact that every interval $[i\eta, (i+1)\eta) \subset [\zeta_j, \zeta_{j+1})$ for some $j \in \{0, 1, \cdots, N(T)\}$, we know that the control $\alpha$ and $\widetilde{\alpha}$ are the same in the interval $[i\eta, (i+1)\eta]$ and the diffusion matrix $\Sigma$ is just constant matrix. Thus, we know that matrix $\Sigma^{-1}(\widetilde{\alpha}(i\eta)) = \Sigma^{-1}(\alpha(i\eta))$, which takes one of the form from $(\Sigma^{-1}(0), \Sigma^{-1}(1)) :=$
$\left\{ \left( \begin{matrix} \frac{1}{\sqrt{2\tau^{(1)}}} \mathbf{I}_d & 0 \\ 0 & \frac{1}{\sqrt{2\tau^{(2)}}} \mathbf{I}_d \end{matrix} \right), \left( \begin{matrix} \frac{1}{\sqrt{2\tau^{(2)}}} \mathbf{I}_d & 0 \\ 0 & \frac{1}{\sqrt{2\tau^{(1)}}} \mathbf{I}_d \end{matrix} \right) \right\}$. If $\Sigma^{-1}(\alpha(i\eta)) = \Sigma^{-1}(0)$, we get

$$
\begin{aligned}
& \Big\| \Sigma^{-1}(\widetilde{\alpha}(i\eta)) \nabla \widetilde{G}(\boldsymbol{\beta}_t) - \Sigma^{-1}(\alpha(i\eta)) \nabla G(\boldsymbol{\beta}_t) \Big\|^2 \\
=& \sum_{j=1}^{d} \frac{1}{2\tau^{(1)}} |\nabla_j \widetilde{G}(\boldsymbol{\beta}_t) - \nabla_j G(\boldsymbol{\beta}_t)|^2 + \sum_{j=d+1}^{2d} \frac{1}{2\tau^{(2)}} |\nabla_j \widetilde{G}(\boldsymbol{\beta}_t) - \nabla_j G(\boldsymbol{\beta}_t)|^2 \\
\leq& \frac{1}{2\tau^{(1)}} \sum_{j=1}^{2d} |\nabla_j \widetilde{G}(\boldsymbol{\beta}_t) - \nabla_j G(\boldsymbol{\beta}_t)|^2 \leq \frac{1}{2\tau^{(1)}} \| \nabla \widetilde{G}(\boldsymbol{\beta}_t) - \nabla_j G(\boldsymbol{\beta}_t) \|^2.
\end{aligned}
$$

Here $\nabla G(\boldsymbol{\beta}) := \left( \begin{matrix} \nabla L(\boldsymbol{\beta}^{(1)}) \\ \nabla L(\boldsymbol{\beta}^{(2)}) \end{matrix} \right)$ and $\nabla \widetilde{G}(\boldsymbol{\beta}) := \left( \begin{matrix} \nabla \widetilde{L}(\boldsymbol{\beta}^{(1)}) \\ \nabla \widetilde{L}(\boldsymbol{\beta}^{(2)}) \end{matrix} \right)$. The other matrix form of $\Sigma^{-1}(1)$ will result in the same estimates. We thus get

$$
\mathcal{I} \leq \frac{1}{4\tau^{(1)}} \sum_{i=0}^{k-1} \mathbb{E}_{\nu_T, \boldsymbol{\beta}} \Big[ \int_{i\eta}^{(i+1)\eta} \Big\| \nabla \widetilde{G}(\boldsymbol{\beta}_t) - \nabla G(\boldsymbol{\beta}_t) \Big\|^2 dt \Big]
$$

On each fixed interval, for $t \in [k\eta, (k+1)\eta)$, we have $\mathbb{P}^G$-Brownian motion and $\mathbb{P}^{\widetilde{G}}$-Brownian motion (see examples in Theorem 8.6.6 and Example 8.6.9 (Øksendal, 2003)),

$$
\begin{aligned}
d\boldsymbol{W}_t^G &= d\boldsymbol{W}_t + \Sigma^{-1}(\alpha_t)(\nabla G(\boldsymbol{\beta}_t)) dt. \\
d\boldsymbol{W}_t^{\widetilde{G}} &= d\boldsymbol{W}_t + \Sigma^{-1}(\alpha_t)(\nabla \widetilde{G}(\widetilde{\boldsymbol{\beta}}_t)) dt.
\end{aligned}
$$

Plugging the $\mathbb{P}^G$ (and $\mathbb{P}^{\widetilde{G}}$)-Brownian motions to the original dynamics (34) and (35), we have

$$
d\boldsymbol{\beta}_t = \Sigma(\alpha_t) d\boldsymbol{W}_t^G, \quad \text{and} \quad d\widetilde{\boldsymbol{\beta}}_t = \Sigma(\alpha_t) d\boldsymbol{W}_t^{\widetilde{G}}.
$$

On each interval $[i\eta, (i+1)\eta)$, $\Sigma(\alpha_t)$ is a constant matrix, thus we know that the probability distribution of $\{\beta_t\}_{t \in [k\eta, (k+1)\eta)}$ and $\{\widetilde{\beta}_t\}_{t \in [k\eta, (k+1)\eta)}$ are the same and we denote as $\mathcal{L}(\beta_t) = \mathcal{L}(\widetilde{\beta}_t)$. The difference is that $\beta_t$ is driven by $\mathbb{P}^G$-Brownian motion and $\widetilde{\beta}_t$ is driven by $\mathbb{P}^{\widetilde{G}}$-Brownian motion, which implies that, for $t \in [i\eta, (i+1)\eta)$, we have

$$\mathbb{E}_{\nu_T, \beta}\Big[\Big\|\nabla\widetilde{G}(\beta_t) - \nabla G(\beta_t)\Big\|^2\Big] = \mathbb{E}_{\mu_T, \widetilde{\beta}}\Big[\Big\|\nabla\widetilde{G}(\widetilde{\beta}_t) - \nabla G(\widetilde{\beta}_t)\Big\|^2\Big]. \tag{42}$$

Thus, we have the following estimates,

$$
\begin{aligned}
\mathcal{I} \leq & \frac{1}{4\tau^{(1)}}\sum_{i=0}^{k-1}\mathbb{E}_{\mu_T, \widetilde{\beta}}\Big[\int_{i\eta}^{(i+1)\eta}\Big\|\nabla G(\widetilde{\beta}_t) - \nabla G(\widetilde{\beta}_{\lfloor t/\eta\rfloor\eta})\Big\|^2 dt\Big] \\
& + \frac{1}{4\tau^{(1)}}\sum_{i=0}^{k-1}\mathbb{E}_{\mu_T, \widetilde{\beta}}\Big[\int_{i\eta}^{(i+1)\eta}\Big\|\nabla G(\widetilde{\beta}_{\lfloor t/\eta\rfloor\eta}) - \nabla\widetilde{G}(\widetilde{\beta}_{\lfloor t/\eta\rfloor\eta})\Big\|^2 dt\Big] \\
\leq & \frac{C^2}{4\tau^{(1)}}\sum_{i=0}^{k-1}\mathbb{E}_{\mu_T, \widetilde{\beta}}\Big[\int_{i\eta}^{(i+1)\eta}\Big\|\widetilde{\beta}_t - \widetilde{\beta}_{i\eta})\Big\|^2 dt\Big] \cdots \mathcal{I}_1 \\
& + \frac{1}{4\tau^{(1)}}\sum_{i=0}^{k-1}\mathbb{E}_{\mu_T, \widetilde{\beta}}\Big[\int_{i\eta}^{(i+1)\eta}\Big\|\nabla G(\widetilde{\beta}_{\lfloor t/\eta\rfloor\eta}) - \nabla\widetilde{G}(\widetilde{\beta}_{\lfloor t/\eta\rfloor\eta})\Big\|^2 dt\Big] \cdots \mathcal{I}_2.
\end{aligned}
$$

We now estimate the two terms $\mathcal{I}_1$ and $\mathcal{I}_2$ separately. Notice that, following our notation of $\mathbb{P}^{\widetilde{G}}$-Brownian motion, for $t \in [i\eta, (i+1)\eta)$, we have

$$\widetilde{\beta}_t - \widetilde{\beta}_{i\eta} = \Sigma(\alpha_t)(W_t^{\widetilde{G}} - W_{i\eta}^{\widetilde{G}}) = \Sigma(\alpha_t)(W_t^{\widetilde{G}} - W_{i\eta}^{\widetilde{G}}),$$

which implies that (recall that $d\mu_T = d\mathbb{P}^{\widetilde{G}} \times \mathbf{N}^{\widetilde{S}}$ and $\Sigma \in \mathbb{R}^{2d \times 2d}$),

$$\mathbb{E}_{\mu_T, \widetilde{\beta}}[\|\widetilde{\beta}_t - \widetilde{\beta}_{i\eta}\|^2] \leq 2\tau^{(1)}d\eta + 2\tau^{(2)}d\eta \leq 4\tau^{(2)}d\eta.$$

We thus conclude that,

$$\mathcal{I}_1 \leq C^2\frac{\tau^{(2)}}{\tau^{(1)}}kd\eta^2.$$

As for the term $\mathcal{I}_2$, according to Assumption 3, we obtain that

$$\mathcal{I}_2 \leq \frac{\eta\delta}{4\tau^{(1)}}\sum_{i=0}^{k-1}\mathbb{E}_{\mu_T, \widetilde{\beta}}\Big[C^2\|\widetilde{\beta}_{i\eta}\|^2 + \Phi^2\Big].$$

Now, we just need to estimate $E_{\mu_T, \widetilde{\beta}}[\|\widetilde{\beta}_{k\eta}\|^2]$ [†]. On each interval $[i\eta, (i+1)\eta]$, under the measure $d\mu_{T, \widetilde{\beta}}$, we have

$$\widetilde{\beta}_{(i+1)\eta} = \widetilde{\beta}_{i\eta} + \Sigma(\alpha(i\eta))(W_{(i+1)\eta}^{\widetilde{G}} - W_{i\eta}^{\widetilde{G}}),$$

which implies that

$$
\begin{aligned}
& \mathbb{E}_{\mu_T, \widetilde{\beta}}[\|\widetilde{\beta}_{(i+1)\eta}\|^2] \\
= & \mathbb{E}_{\mu_T, \widetilde{\beta}}[\|\widetilde{\beta}_{i\eta}\|^2] + \mathbb{E}_{\mu_T, \widetilde{\beta}}[\langle\widetilde{\beta}_{i\eta}, W_{(i+1)\eta}^{\widetilde{G}} - W_{i\eta}^{\widetilde{G}}\rangle] + \mathbb{E}_{\mu_T, \widetilde{\beta}}[\|W_{(i+1)\eta}^{\widetilde{G}} - W_{i\eta}^{\widetilde{G}}\|^2] \\
= & \mathbb{E}_{\mu_T, \widetilde{\beta}}[\|\widetilde{\beta}_{i\eta}\|^2] + [2\tau^{(1)} + 2\tau^{(2)}]d\eta
\end{aligned}
$$

The last equality follows from the independence of $\widetilde{\beta}_{k\eta}$ and $W_{(k+1)\eta}^{\widetilde{G}} - W_{k\eta}^{\widetilde{G}}$, and $W^{\widetilde{G}}$ is a $\mathbb{P}^{\widetilde{G}}$-Brownian motion. By induction, we get

$$\mathbb{E}_{\mu_T, \widetilde{\beta}}[\|\widetilde{\beta}_{i\eta}\|^2] \leq 2id[\tau^{(1)} + \tau^{(2)}]\eta \leq 2kd[\tau^{(1)} + \tau^{(2)}].$$

---

[†]In principle, the Wiener measure $W$ under $\mathbb{P}^{\widetilde{G}}$ is not a Brownian motion, thus the uniform $L^2$ bound used in Lemma.3 may not be appropriate. Instead, we estimate the upper bound using a slightly weaker result.

We conclude that,

$$\mathcal{I}_2 \leq \frac{k\eta\delta}{4\tau^{(1)}}\Big(2C^2[\tau^{(1)} + \tau^{(2)}]kd\eta + \Phi^2\Big),$$

which implies that

$$\mathcal{I} \leq \frac{k\eta}{4\tau^{(1)}}\Big(2\delta C^2[\tau^{(1)} + \tau^{(2)}]kd\eta + \delta\Phi^2\Big) + C^2\frac{\tau^{(2)}}{\tau^{(1)}}kd\eta^2.$$

**Estimate $\mathcal{J}$:** According to our definition of the swapping probability, we have, for each $i$,

$$\widetilde{S}(\widetilde{\boldsymbol{\beta}}_{i\eta}) = \min\{1, \widetilde{S}_{\eta,m,n}(\widetilde{\boldsymbol{\beta}}_{i\eta}^{(1)}, \widetilde{\boldsymbol{\beta}}_{i\eta}^{(2)})\}, \quad S(\boldsymbol{\beta}_{i\eta}) = \min\{1, S(\boldsymbol{\beta}_{i\eta}^{(1)}, \boldsymbol{\beta}_{i\eta}^{(2)})\},$$

which means $|\widetilde{S}(\widetilde{\boldsymbol{\beta}}_{i\eta}) - S(\boldsymbol{\beta}_{i\eta})| \leq 1$. Denote $C_\tau = |\frac{1}{\tau^{(1)}} - \frac{1}{\tau^{(2)}}|$, we have

$$\widetilde{S}_{\eta,m,n}(\widetilde{\boldsymbol{\beta}}_{i\eta}^{(1)}, \widetilde{\boldsymbol{\beta}}_{i\eta}^{(2)}) = \exp\Big(C_\tau(\widetilde{L}(B_{i\eta}|\boldsymbol{\beta}_{i\eta}^{(1)}) - \widetilde{L}(B_{i\eta}|\boldsymbol{\beta}_{i\eta}^{(2)})) - C_\tau^2\frac{\widetilde{\sigma}^2}{2}\Big)$$

$$S(\boldsymbol{\beta}_{i\eta}^{(1)}, \boldsymbol{\beta}_{i\eta}^{(2)}) = \exp\Big(C_\tau(L(\boldsymbol{\beta}_{i\eta}^{(1)}) - L(\boldsymbol{\beta}_{i\eta}^{(2)}))\Big).$$

Applying Taylor expansion for the exponential function at $C_\tau(L(\boldsymbol{\beta}_{k\eta}^{(1)}) - L(\boldsymbol{\beta}_{k\eta}^{(2)}))$, we have

$$\mathbb{E}_{\nu_T,\boldsymbol{\beta}}\Big[|\widetilde{S}_{\eta,m,n}(\widetilde{\boldsymbol{\beta}}_{i\eta}^{(1)}, \widetilde{\boldsymbol{\beta}}_{i\eta}^{(2)}) - S(\boldsymbol{\beta}_{i\eta}^{(1)}, \boldsymbol{\beta}_{i\eta}^{(2)})|\Big]$$

$$\lesssim \mathbb{E}_{\nu_T,\boldsymbol{\beta}}\Big[S(\boldsymbol{\beta}_{i\eta}^{(1)}, \boldsymbol{\beta}_{i\eta}^{(2)})\Big|C_\tau(\widetilde{L}(B_{k\eta}|\boldsymbol{\beta}_{i\eta}^{(1)}) - \widetilde{L}(B_{k\eta}|\boldsymbol{\beta}_{i\eta}^{(2)})) - C_\tau^2\frac{\widetilde{\sigma}^2}{2} - C_\tau(L(\boldsymbol{\beta}_{i\eta}^{(1)}) - L(\boldsymbol{\beta}_{i\eta}^{(2)}))\Big| + \text{higher order term}\Big]$$

$$\leq \mathbb{E}_{\nu_T,\boldsymbol{\beta}}\Big[\Big|C_\tau(\widetilde{L}(B_{i\eta}|\boldsymbol{\beta}_{i\eta}^{(1)}) - \widetilde{L}(B_{i\eta}|\boldsymbol{\beta}_{i\eta}^{(2)})) - C_\tau^2\frac{\widetilde{\sigma}^2}{2} - C_\tau(L(\boldsymbol{\beta}_{i\eta}^{(1)}) - L(\boldsymbol{\beta}_{i\eta}^{(2)}))\Big| + \mathcal{O}(\widetilde{\sigma}^2)\Big]$$

where the last inequality follows from $S(\boldsymbol{\beta}_{i\eta}^{(1)}, \boldsymbol{\beta}_{i\eta}^{(2)}) \leq 1$. Combining Lemma B1, we thus get the following estimates,

$$\mathcal{J} = \sum_{i=0}^{k-1} \mathbb{E}_{\nu_T,\boldsymbol{\beta}}\Big[r|\widetilde{S}_{\eta,m,n}(\widetilde{\boldsymbol{\beta}}_{i\eta}) - S(\boldsymbol{\beta}_{i\eta})|\eta\Big]$$

$$\leq r\eta \sum_{i=0}^{k-1} \mathbb{E}_{\nu_T,\boldsymbol{\beta}}\Big[\Big|C_\tau(\widetilde{L}(B_{i\eta}|\boldsymbol{\beta}_{i\eta}^{(1)}) - \widetilde{L}(B_{i\eta}|\boldsymbol{\beta}_{i\eta}^{(2)})) - C_\tau^2\frac{\widetilde{\sigma}^2}{2} - C_\tau(L(\boldsymbol{\beta}_{i\eta}^{(1)}) - L(\boldsymbol{\beta}_{i\eta}^{(2)}))\Big| + \mathcal{O}(\widetilde{\sigma}^2)\Big]$$

$$\leq rk\eta\mathcal{O}(C_\tau\widetilde{\sigma} + \widetilde{\sigma}^2) = rk\eta\mathcal{O}\left(\left(\frac{m^2}{n}\eta\right)^{1/2} d\right)$$

where the last inequality follows from the Jensen's inequality and the last order holds given a large enough batch size $n$ or a small enough $m$ and $\eta$.

**Estimate $\mathcal{K}$:** We now estimate the last term $\mathcal{K}$, we have

$$\mathcal{K} = \sum_{j=0}^{N(T)} \mathbb{E}_{\nu_T,\boldsymbol{\beta}}\Big[|\log\widetilde{S}_{\eta,m,n}(\widetilde{\boldsymbol{\beta}}_{\zeta_j}) - \log S(\boldsymbol{\beta}_{\zeta_j})|\Big]$$

$$\leq C_\tau \sum_{j=0}^{N(T)} \mathbb{E}_{\nu_T,\boldsymbol{\beta}}\Big[\Big|[\widetilde{L}(B_{\zeta_j}|\boldsymbol{\beta}_{\zeta_j}^{(1)}) - \widetilde{L}(B_{\zeta_j}|\boldsymbol{\beta}_{\zeta_j}^{(2)}) - C_\tau\frac{\widetilde{\sigma}^2}{2}] - [L(\boldsymbol{\beta}_{\zeta_j}^{(1)}) - L(\boldsymbol{\beta}_{\zeta_j}^{(2)})]\Big|\Big]$$

$$\leq N(T)C_\tau^2\mathbb{E}_{\nu_T,\boldsymbol{\beta}}[\widetilde{\sigma}^2/2] + C_\tau \sum_{j=1}^{N(T)} \text{Var}[\widetilde{L}(B_{\zeta_j}|\boldsymbol{\beta}_{\zeta_j}^{(1)}) - \widetilde{L}(B_{\zeta_j}|\boldsymbol{\beta}_{\zeta_j}^{(2)})]^{1/2}$$

$$\leq \frac{N(T)C_\tau^2\widetilde{\sigma}^2}{2} + N(T)C_\tau\widetilde{\sigma}$$

Combining Lemma B1 again, we conclude with

$$\mathcal{K} \leq C_\tau^2 \frac{N(T)\widetilde{\sigma}^2}{2} + N(T)C_\tau\widetilde{\sigma} = N(T)\mathcal{O}\left(\left(\frac{m^2}{n}\eta\right)^{1/2} d\right).$$

Combining the estimates of $\mathcal{I}$, $\mathcal{J}$, and $\mathcal{K}$, we complete the proof.

**Remark 3** *After the change of measure, the expectation is under the new measure $\mathbb{P}^G$ (or $\mathbb{P}^{\widetilde{G}}$) instead of the Wiener measure $\mathbb{P}$. In the estimate of term $\mathcal{I}$, similar $L^2$ estimates of the term $\mathbb{E}_{\mu_T,\widetilde{\beta}}[\|\widetilde{\beta}_{(i+1)\eta}\|^2]$ has been obtained in Raginsky et al. (2017)[Proof of Lemma 7] when there is no swap. The difference is we write the dynamic of $\widetilde{\beta}_{(i+1)\eta}$ with respect to the $\mathbb{P}^{\widetilde{G}}$-Brownian motion $W^{\widetilde{G}}$ instead of the $\mathbb{P}$-Brownian motion $W$. In principle, $W$ under $\mathbb{P}^{\widetilde{G}}$ is not a Brownian motion.*

We then extend the distance of relative entropy $D_{KL}(\mu_T|\nu_T)$ to the Wasserstein distance $\mathcal{W}_2(\mu_T, \nu_T)$ via a weighted transportation-cost inequality of Bolley & Villani (2005).

**Theorem 2** *Given a large enough batch size $n$ or a small enough $m$ and $\eta$, we have*

$$\mathcal{W}_2(\mu_T, \nu_T) \leq \mathcal{O}\left(dk^{3/2}\eta\left(\eta^{1/4} + \delta^{1/4} + \left(\frac{m^2}{n}\eta\right)^{1/8}\right)\right). \tag{43}$$

**Proof** Before we proceed, we first show in Lemma.D5 that $\nu_T$ has a bounded second moment; the $L_2$ upper bound of $\mu_T$ is majorly proved in Lemma.C2 (Chen et al., 2019) except that the slight difference is that the constant in the RHS of (C.38) Chen et al. (2019) is changed to account for the stochastic noise. Then applying Corollary 2.3 in Bolley & Villani (2005), we can upper bound the two Borel probability measures $\mu_T$ and $\nu_T$ with finite second moments as follows

$$\mathcal{W}_2(\mu_T, \nu_T) \leq C_\nu\left[\sqrt{D_{KL}(\mu_T|\nu_T)} + \left(\frac{D_{KL}(\mu_T|\nu_T)}{2}\right)^{1/4}\right], \tag{44}$$

where $C_\nu = 2\inf_{\lambda>0}\left(\frac{1}{\lambda}\left(\frac{3}{2} + \log\int_{\mathbb{R}^d}e^{\lambda\|w\|^2}\nu(dw)\right)\right)^{1/2}$. Applying Lemma D6, we have

$$\mathcal{W}_2^2(\mu_T, \nu_T) \leq \left(12 + 8\left(\kappa_0 + 2b + 4d\tau^{(2)}\right)k\eta\right)\left(D_{KL}(\mu_T|\nu_T) + \sqrt{D_{KL}(\mu_T|\nu_T)}\right).$$

Combining Lemma.C2 and $\sqrt{N(T)} \leq N(T)$ and taking $\eta \leq 1$, $k\eta > 1$, and $\lambda = 1$, we have

$$\mathcal{W}_2^2(\mu_{T,\widetilde{\beta}}, \nu_{T,\beta}) \leq \left(12 + 8\left(\kappa_0 + 2b + 4d\tau^{(2)}\right)\right)k\eta\left((\widetilde{\Phi}_0 + \widetilde{\Phi}_1\sqrt{\eta})k\eta + N(T)\widetilde{\Phi}_2\right),$$

where $\widetilde{\Phi}_i = \Phi_i + \sqrt{\Phi_i}$ for $i \in \{0, 1, 2\}$. In what follows, we have

$$\mathcal{W}_2^2(\mu_{T,\widetilde{\beta}}, \nu_{T,\beta}) \leq \left(\Psi_0 + \Psi_1\sqrt{\eta}\right)(k\eta)^2 + \Psi_2 k\eta N(T),$$

where $\Psi_i = \left(12 + 8\left(\kappa_0 + 2b + 4d\tau^{(2)}\right)\right)\widetilde{\Phi}_i$ for $i \in \{0, 1, 2\}$.

By the orders of $\Phi_0$, $\Phi_1$ and $\Phi_2$ defined in Lemma.C2, we have

$$\mathcal{W}_2^2(\mu_{T,\widetilde{\beta}}, \nu_{T,\beta}) \leq \mathcal{O}\left(d^2k^3\eta^2\left(\eta^{1/2} + \delta^{1/2} + \left(\frac{m^2}{n}\eta\right)^{1/4} + \frac{N(T)}{k\eta}\left(\frac{m^2}{n}\eta\right)^{1/4}\right)\right)$$

$$\leq \mathcal{O}\left(d^2k^3\eta^2\left(\eta^{1/2} + \delta^{1/2} + \left(\frac{m^2}{n}\eta\right)^{1/4}\right)\right),$$

where $\frac{N(T)}{k\eta}$ can be interpreted as the average swapping rate from time $0$ to $T$ and is of order $\mathcal{O}(1)$. Taking square root to both sides of the above inequality lead to the desired result (43).

## D  PROOF OF TECHNICAL LEMMAS

**Lemma D1 (Local Lipschitz continuity)** *Given a $d$-dimensional centered ball $U$ of radius $R$, $L(\cdot)$ is $D_R$-Lipschitz continuous in that $|L(\mathbf{x}_i|\boldsymbol{\beta}_1) - L(\mathbf{x}_i|\boldsymbol{\beta}_2)| \leq \frac{D_R}{N}\|\boldsymbol{\beta}_1 - \boldsymbol{\beta}_2\|$ for $\forall \boldsymbol{\beta}_1, \boldsymbol{\beta}_2 \in U$ and any $i \in \{1, 2, \cdots, N\}$, where $D_R = CR + \max_{i \in \{1,2,\cdots,N\}} N\|\nabla L(\mathbf{x}_i|\boldsymbol{\beta}_\star)\| + \frac{Cb}{a}$.*

**Proof**

For any $\boldsymbol{\beta}_1, \boldsymbol{\beta}_2 \in U$, there exists $\boldsymbol{\beta}_3 \in U$ that satisfies the mean-value theorem such that

$$|L(\mathbf{x}_i|\boldsymbol{\beta}_1) - L(\mathbf{x}_i|\boldsymbol{\beta}_2)| = \langle \nabla L(\mathbf{x}_i|\boldsymbol{\beta}_3), \boldsymbol{\beta}_1 - \boldsymbol{\beta}_2 \rangle \leq \|\nabla L(\mathbf{x}_i|\boldsymbol{\beta}_3)\| \cdot \|\boldsymbol{\beta}_1 - \boldsymbol{\beta}_2\|,$$

Moreover, by Lemma D2, we have

$$|L(\mathbf{x}_i|\boldsymbol{\beta}_1) - L(\mathbf{x}_i|\boldsymbol{\beta}_2)| \leq \|\nabla L(\mathbf{x}_i|\boldsymbol{\beta}_3)\| \cdot \|\boldsymbol{\beta}_1 - \boldsymbol{\beta}_2\| \leq \frac{CR + Q}{N}\|\boldsymbol{\beta}_1 - \boldsymbol{\beta}_2\|. \blacksquare$$

**Lemma D2** *Under the smoothness and dissipativity assumptions 1, 2, for any $\boldsymbol{\beta} \in \mathbb{R}^d$, it follows that*

$$\|\nabla L(\mathbf{x}_i|\boldsymbol{\beta})\| \leq \frac{C}{N}\|\boldsymbol{\beta}\| + \frac{Q}{N}. \tag{45}$$

*where $Q = \max_{i \in \{1,2,\cdots,N\}} N\|\nabla L(\mathbf{x}_i|\boldsymbol{\beta}_\star)\| + \frac{bC}{a}$.*

**Proof**  According to the dissipativity assumption, we have

$$\langle \boldsymbol{\beta}_\star, \nabla L(\boldsymbol{\beta}_\star) \rangle \geq a\|\boldsymbol{\beta}^\star\|^2 - b, \tag{46}$$

where $\boldsymbol{\beta}_\star$ is a minimizer of $\nabla L(\cdot)$ such that $\nabla L(\boldsymbol{\beta}_\star) = 0$. In what follows, we have $\|\boldsymbol{\beta}_\star\| \leq \frac{b}{a}$.

Combining the triangle inequality and the smoothness assumption 1, we have

$$\|\nabla L(\mathbf{x}_i|\boldsymbol{\beta})\| \leq C_N\|\boldsymbol{\beta} - \boldsymbol{\beta}_\star\| + \|\nabla L(\mathbf{x}_i|\boldsymbol{\beta}_\star)\| \leq C_N\|\boldsymbol{\beta}\| + \frac{C_N b}{a} + \|\nabla L(\mathbf{x}_i|\boldsymbol{\beta}_\star)\|. \tag{47}$$

Setting $C_N = \frac{C}{N}$ as in (11) and $Q = \max_{i \in \{1,2,\cdots,N\}} \|\nabla L(\mathbf{x}_i|\boldsymbol{\beta}_\star)\| + \frac{bC}{a}$ completes the proof.  $\blacksquare$

The following lemma is majorly adapted from Lemma C.2 of Chen et al. (2019), except that the corresponding constant in the RHS of (C.38) is slightly changed to account for the stochastic noise. A similar technique has been established in Lemma 3 of Raginsky et al. (2017).

**Lemma D3 (Uniform $L^2$ bounds on replica exchange SGLD)** *Under the smoothness and dissipativity assumptions 1, 2. Given a small enough learning rate $\eta \in (0, 1 \vee \frac{a}{C^2})$, there exists a positive constant $\Psi_{d,\tau^{(2)},C,a,b} < \infty$ such that $\sup_{k \geq 1} \mathbb{E}[\|\boldsymbol{\beta}_k\|^2] < \Psi_{d,\tau^{(2)},C,a,b}$.*

**Lemma D4 (Exponential dependence on the variance)** *Assume $S$ is a log-normal distribution with mean $u - \frac{1}{2}\sigma^2$ and variance $\sigma^2$ on the log scale. Then $\mathbb{E}[\min(1, S)] = \mathcal{O}(e^{u - \frac{\sigma^2}{8}})$, which is exponentially smaller given a large variance $\sigma^2$.*

**Proof**  For a log-normal distribution $S$ with mean $u - \frac{1}{2}\sigma^2$ and variance $\sigma^2$ on the log scale, the probability density $f_S(S)$ follows that $\frac{1}{S\sqrt{2\pi\sigma^2}} \exp\left\{-\frac{(\log S - u + \frac{1}{2}\sigma^2)^2}{2\sigma^2}\right\}$. In what follows, we have

$$\mathbb{E}[\min(1, S)] = \int_0^\infty \min(1, S) f_S(S) dS = \int_0^\infty \min(1, S) \frac{1}{S\sqrt{2\pi\sigma^2}} \exp\left\{-\frac{(\log S - u + \frac{1}{2}\sigma^2)^2}{2\sigma^2}\right\} dS$$

By change of variable $y = \frac{\log S - u + \frac{1}{2}\sigma^2}{\sigma}$ where $S = e^{\sigma y + u - \frac{1}{2}\sigma^2}$ and $y = -\frac{u}{\sigma} + \frac{\sigma}{2}$ given $S = 1$, it follows that

$$
\begin{aligned}
&\mathbb{E}[\min(1, S)] \\
&= \int_0^1 S \frac{1}{S\sqrt{2\pi\sigma^2}} \exp\left\{-\frac{(\log S - u + \frac{1}{2}\sigma^2)^2}{2\sigma^2}\right\} dS + \int_1^\infty \frac{1}{S\sqrt{2\pi\sigma^2}} \exp\left\{-\frac{(\log S - u + \frac{1}{2}\sigma^2)^2}{2\sigma^2}\right\} dS \\
&= \int_{-\infty}^{-\frac{u}{\sigma} + \frac{\sigma}{2}} \frac{1}{\sqrt{2\pi\sigma^2}} e^{-\frac{y^2}{2}} \sigma e^{u - \frac{1}{2}\sigma^2 + \sigma y} dy + \int_{-\frac{u}{\sigma} + \frac{\sigma}{2}}^\infty \frac{1}{\sqrt{2\pi\sigma^2}} e^{-\sigma y - u + \frac{1}{2}\sigma^2} e^{-\frac{y^2}{2}} \sigma e^{u - \frac{1}{2}\sigma^2 + \sigma y} dy \\
&= e^u \int_{-\infty}^{-\frac{u}{\sigma} + \frac{\sigma}{2}} \frac{1}{\sqrt{2\pi}} e^{-\frac{(y - \sigma)^2}{2}} dy + \frac{1}{\sigma} \int_{-\frac{u}{\sigma} + \frac{\sigma}{2}}^\infty \frac{1}{\sqrt{2\pi}} e^{-\frac{y^2}{2}} dy \\
&= e^u \int_{\frac{u}{\sigma} + \frac{\sigma}{2}}^\infty \frac{1}{\sqrt{2\pi}} e^{-\frac{z^2}{2}} dz + \frac{1}{\sigma} \int_{-\frac{u}{\sigma} + \frac{\sigma}{2}}^\infty \frac{1}{\sqrt{2\pi}} e^{-\frac{y^2}{2}} dy \\
&\leq e^u \int_{-\frac{u}{\sigma} + \frac{\sigma}{2}}^\infty \frac{1}{\sqrt{2\pi}} e^{-\frac{z^2}{2}} dz + \frac{1}{\sigma} \int_{-\frac{u}{\sigma} + \frac{\sigma}{2}}^\infty \frac{1}{\sqrt{2\pi}} e^{-\frac{y^2}{2}} dy \\
&\leq \left(e^u + \frac{1}{\sigma}\right) e^{-\frac{(-\frac{u}{\sigma} + \frac{\sigma}{2})^2}{2}} \lesssim e^{u - \frac{\sigma^2}{8}},
\end{aligned}
$$

where the last equality follows from the change of variable $z = \sigma - y$ and the second last inequality follows from the exponential tail bound of the standard Gaussian variable $\mathbb{P}(y > \epsilon) \leq e^{\frac{-\epsilon^2}{2}}$. ∎

**Lemma D5 (Uniform $L^2$ bound on replica exchange Langevin diffusion)** *For all $\eta \in (0, 1 \wedge \frac{a}{4C^2})$, we have that*

$$
\mathbb{E}[\|(\boldsymbol{\beta}_t^{(1)}, \boldsymbol{\beta}_t^{(2)})\|^2] \leq \mathbb{E}[e^{\|\boldsymbol{\beta}_0^{(1)}, \boldsymbol{\beta}_0^{(2)}\|^2}] + \frac{b + 2d\tau^{(2)}}{a}.
$$

**Proof** Consider $L_t(\boldsymbol{\beta}_t) = \|\boldsymbol{\beta}_t\|^2$, where $\boldsymbol{\beta}_t = (\boldsymbol{\beta}_t^{(1)}, \boldsymbol{\beta}_t^{(2)}) \in \mathbb{R}^{2d}$. The proof is marjorly adapted from Lemma 3 in Raginsky et al. (2017), except that the generalized Itô formula (formula 2.7 in page 29 of Yin & Zhu (2010)) is used to handle the jump operator, which follows that

$$
\begin{aligned}
dL_t = &-2\langle \boldsymbol{\beta}_t, \nabla G(\boldsymbol{\beta}_t)\rangle + 2d(\tau^{(1)} + \tau^{(2)})dt + 2\boldsymbol{\beta}_t^T \Sigma(\alpha_t) dW(t) \\
&+ \underbrace{r S_{\eta, m, n}(\boldsymbol{\beta}_t^{(1)}, \boldsymbol{\beta}_t^{(2)}) \cdot (L_t(\boldsymbol{\beta}^{(2)}, \boldsymbol{\beta}^{(1)}) - L_t(\boldsymbol{\beta}_t^{(1)}, \boldsymbol{\beta}_t^{(2)}))}_{\text{Jump-inducing drift}} + M_1(t) + M_2(t),
\end{aligned}
$$

where $\nabla G(\boldsymbol{\beta}) := \begin{pmatrix} \nabla L(\boldsymbol{\beta}^{(1)}) \\ \nabla L(\boldsymbol{\beta}^{(2)}) \end{pmatrix}$ and $M_1(t)$ and $M_2(t)$ are two martingales defined in formula 2.7 in Yin & Zhu (2010)). Due to the definition of $L_t(\boldsymbol{\beta}_t)$, we have $L_t(\boldsymbol{\beta}_t^{(1)}, \boldsymbol{\beta}_t^{(2)}) = L_t(\boldsymbol{\beta}_t^{(2)}, \boldsymbol{\beta}_t^{(1)})$, which implies that the Jump-inducing drift actually disappears. Taking expectations and applying the margingale property of the Itô integral, we have the almost the same upper bound as Lemma 3 in Raginsky et al. (2017). Combining $\mathbb{E}[\|\boldsymbol{\beta}_0\|^2] \leq \log \mathbb{E}[e^{\|\boldsymbol{\beta}_0\|^2}]$ completes the proof.

**Lemma D6 (Exponential integrability of replica exchange Langevin diffusion)** *For all $\tau \leq \frac{2}{a}$, it follows that*

$$
\log \mathbb{E}[e^{\|(\boldsymbol{\beta}_t^{(1)}, \boldsymbol{\beta}_t^{(2)})\|^2}] \leq \underbrace{\log \mathbb{E}[e^{\|(\boldsymbol{\beta}_0^{(1)}, \boldsymbol{\beta}_0^{(2)})\|^2}]}_{\kappa_0} + 2(b + 2d\tau^{(2)})t.
$$

**Proof** The proof is marjorly adapted from Lemma 4 in Raginsky et al. (2017). The only difference is that the generalized Itô formula (formula 2.7 in Yin & Zhu (2010)) is used again as in Lemma D5. Consider $L(t, \boldsymbol{\beta}_t) = e^{\|\boldsymbol{\beta}_t\|^2}$, where $\boldsymbol{\beta} = (\boldsymbol{\beta}_t^{(1)}, \boldsymbol{\beta}_t^{(2)}) \in \mathbb{R}^{2d}$. Due to the special structure that $L(t, \boldsymbol{\beta}_t)$ is invariant under the swaps of $(\boldsymbol{\beta}_t^{(1)}, \boldsymbol{\beta}_t^{(2)})$, the generator of $L(t, \boldsymbol{\beta}_t)$ with swaps is the same as the one without swaps. Therefore, the desired result follows directly by repeating the steps from Lemma 4 in Raginsky et al. (2017).

---

**Algorithm 2** Adaptive variance-reduced replica exchange SGLD. The learning rate and temperature can be set to dynamic to speed up the computations. A larger smoothing factor $\gamma$ captures the trend better but becomes less robust.

---

**Input** Initial parameters $\boldsymbol{\beta}_0^{(1)}$ and $\boldsymbol{\beta}_0^{(2)}$, learning rate $\eta$ and temperatures $\tau^{(1)}$ and $\tau^{(2)}$, correction factor $F$.

**repeat**
  **Parallel sampling** Randomly pick a mini-batch set $B_k$ of size $n$.

$$\boldsymbol{\beta}_k^{(h)} = \boldsymbol{\beta}_{k-1}^{(h)} - \eta \frac{N}{n} \sum_{i \in B_k} \nabla L(\mathbf{x}_i | \boldsymbol{\beta}_{k-1}^{(h)}) + \sqrt{2\eta \tau^{(h)}} \boldsymbol{\xi}_k^{(h)}, \text{ for } h \in \{1, 2\}.$$

  **Variance-reduced energy estimators** Update $\widehat{L}^{(h)} = \sum_{i=1}^N L\left(\mathbf{x}_i | \boldsymbol{\beta}_{m \lfloor \frac{k}{m} \rfloor}^{(h)}\right)$ every $m$ iterations.

$$\widetilde{L}(B_k | \boldsymbol{\beta}_k^{(h)}) = \frac{N}{n} \sum_{i \in B_k} L(\mathbf{x}_i | \boldsymbol{\beta}_k^{(h)}) + \widetilde{c}_k \cdot \left[ \frac{N}{n} \sum_{i \in B_k} L\left(\mathbf{x}_i \Big| \boldsymbol{\beta}_{m \lfloor \frac{k}{m} \rfloor}^{(h)}\right) - \widehat{L}^{(h)} \right], \text{ for } h \in \{1, 2\}.$$

  **if** $k \bmod m = 0$ **then**
    Update $\widetilde{\sigma}_k^2 = (1 - \gamma) \widetilde{\sigma}_{k-m}^2 + \gamma \sigma_k^2$, where $\sigma_k^2$ is an estimate for $\text{Var}\left(\widetilde{L}(B_k | \boldsymbol{\beta}_k^{(1)}) - \widetilde{L}(B_k | \boldsymbol{\beta}_k^{(2)})\right)$.
    Update $\widetilde{c}_k = (1 - \gamma) \widetilde{c}_{k-m} + \gamma c_k$, where $c_k$ is an estimate for $-\dfrac{\text{Cov}\left(L(B | \boldsymbol{\beta}_k^{(h)}), L\left(B | \boldsymbol{\beta}_{m \lfloor \frac{k}{m} \rfloor}^{(h)}\right)\right)}{\text{Var}\left(L\left(B | \boldsymbol{\beta}_{m \lfloor \frac{k}{m} \rfloor}^{(h)}\right)\right)}$.
  **end if**
  **Bias-reduced swaps** Swap $\boldsymbol{\beta}_{k+1}^{(1)}$ and $\boldsymbol{\beta}_{k+1}^{(2)}$ if $u < \widetilde{S}_{\eta, m, n}$, where $u \sim \text{Unif}\,[0, 1]$, and $\widetilde{S}_{\eta, m, n}$ follows

$$\widetilde{S}_{\eta, m, n} = \exp\left\{ \left( \tfrac{1}{\tau^{(1)}} - \tfrac{1}{\tau^{(2)}} \right) \left( \widetilde{L}(B_{k+1} | \boldsymbol{\beta}_{k+1}^{(1)}) - \widetilde{L}(B_{k+1} | \boldsymbol{\beta}_{k+1}^{(2)}) - \tfrac{1}{F} \left( \tfrac{1}{\tau^{(1)}} - \tfrac{1}{\tau^{(2)}} \right) \widetilde{\sigma}_{m \lfloor \frac{k}{m} \rfloor}^2 \right) \right\}.$$

**until** $k = k_{\max}$.
**Output:** $\{\boldsymbol{\beta}_{i\mathbb{T}}^{(1)}\}_{i=1}^{\lfloor k_{\max}/\mathbb{T} \rfloor}$, where $\mathbb{T}$ is the thinning factor.

---

# E  MORE EMPIRICAL STUDY ON IMAGE CLASSIFICATION

## E.1  TRAINING COST

The batch size of $n = 512$ almost doubles the training time and memory, which becomes too costly in larger experiments. A frequent update of control variates using $m = 50$ is even more time-consuming and is not acceptable in practice. The choice of $m$ gives rise to a tradeoff between computational cost and variance reduction. As such, we choose $m = 392$, which still obtains significant reductions of the variance at the cost of 40% increase on the training time. Note that when we set $m = 2000$, the training cost is only increased by 8% while the variance reduction can be still at most 6 times on CIFAR10 and 10 times on CIFAR100.

## E.2  ADAPTIVE COEFFICIENT

We study the correlation coefficient of the noise from the current parameter $\boldsymbol{\beta}_k^{(h)}$, where $h \in \{1, 2\}$, and the control variate $\boldsymbol{\beta}_{m \lfloor \frac{k}{m} \rfloor}^{(h)}$. As shown in Fig.5, the correlation coefficients are only around -0.5 due to the large learning rate in the early period. This implies that VR-reSGHMC may overuse the noise from the control variates and thus fails to fully exploit the potential in variance reduction. In spirit to the adaptive variance, we try the adaptive correlation coefficients to capture the pattern of the time-varying correlation coefficients and present it in Algorithm 2.

As a result, we can further improve the performance of variance reduction by as much as 40% on CIFAR10 and 30% on CIFAR100 in the first 200 epochs. As the training continues and the learning rate decreases, the correlation coefficient is becoming closer to -1. In the late period, there is still 10% improvement compared to the standard VR-reSGHMC.

In a nut shell, we can try adaptive coefficients in the early period when the absolute value of the correlation is lower than 0.5 or just use the vanilla replica exchange stochastic gradient Monte Carlo to avoid the computations of variance reduction.

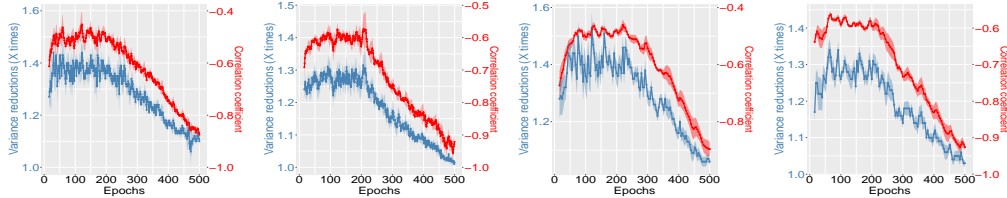

(a) CIFAR10 & m=50    (b) CIFAR100 & m=50    (c) CIFAR10 & m=392    (d) CIFAR100 & m=392

Figure 5: A study of variance reduction techniques using adaptive coefficient and non-adaptive coefficient on CIFAR10 & CIFAR100 datasets.

## F    MORE EMPIRICAL STUDY ON UNCERTAINTY QUANTIFICATION

To avoid sacrificing the prediction power for the known classes, we also include the uncertainty estimate on CIFAR10 using the Brier score (BS) [†] and compare it with the estimates on SVHN. The optimal BS scores on the seen CIFAR10 dataset and the unseen SVHN dataset are 0 and 0.1, respectively. As shown in Table.2, we see that the scores before calibration in the seen CIFAR10 is much lower than the ones in the unseen SVHN. This implies that all the models perform quite well in terms of what it knows, although cSGHMC are slightly better than the alternatives. To alleviate this issue, we propose to calibrate the predictive probability through the temperature scaling (Guo et al., 2017) and obtain much better results. Regarding the BS score on the unseen dataset, we see that M-SGD still performs the worst for frequently making over-confident predictions; SGHMC performs better but is far away from satisfying. reSGHMC obtains much better performance by allowing interactions between different chains. However, the large correction term affects the efficiency of the swaps significantly. In the end, our proposed algorithm increases the efficiency of the swaps via variance reduction and further improves the highly-optimized BS score based on reSGHMC from 0.29 to 0.27, which is much closer to the ideal 0.1. Note that the accurate uncertainty estimates of cVR-reSGHMC on the seen dataset is still maintained. Together with the lowest BS score in the unseen SVHN dataset, cVR-reSGHMC shows its strength in uncertainty quantification.

TABLE 2: UNCERTAINTY ESTIMATES ON SVHN USING CIFAR10 MODELS.

| METHOD | BRIER SCORE (before calibration) | | BRIER SCORE (after calibration) | |
|---|---|---|---|---|
| | CIFAR10 (seen) | SVHN (unseen) | CIFAR10 (seen) | SVHN (unseen) |
| M-SGD | 0.090±0.001 | 0.48±0.02 | 0.098±0.001 | 0.33±0.02 |
| SGHMC | 0.089±0.001 | 0.47±0.02 | 0.099±0.001 | 0.31±0.02 |
| reSGHMC | 0.086±0.002 | 0.41±0.03 | 0.097±0.001 | 0.29±0.02 |
| cSGHMC | 0.084±0.001 | 0.43±0.02 | 0.092±0.001 | 0.30±0.02 |
| cVR-reSGHMC | 0.085±0.001 | 0.38±0.02 | 0.094±0.001 | 0.27±0.02 |

## G    MODIFIED EXAMPLE 5.1

We revisit Example 5.1, and re-run the procedures with temperature $\tau^{(1)} = 1.0$. In Fig. 6, we present trace plots and kernel density estimates (KDE) of samples generated from VR-reSGLD, reSGLD, and SGLD. In particular, we run VR-reSGLD with $m = 40$, $\tau^{(1)} = 1$, $\tau^{(2)} = 500$, $\eta = 1e - 5$, and $F = 1$; reSGLD with the same hyper-parameters as VR-reSGLD except for $F = 500$; and SGLD with $\eta = 1e - 5$ and $\tau = 1$. Note that here, we run reSGLD with a greater $F$ than in Example 5.1 in order to prohibit the drastic reduction of the swapping rate which is caused by the pickier target density. As in Example 5.1, for the ground truth, we run replica exchange Langevin dynamics

---

[†]$BS = \frac{1}{N} \sum_{i=1}^{N} \sum_{j=1}^{R} (f_{ij} - o_{ij})^2$, where $f_i$ is the predictive probability and $o_i$ is actual output of the event which is 1 if it happens and 0 otherwise; $N$ is the number of instances and $R$ is the number of classes.

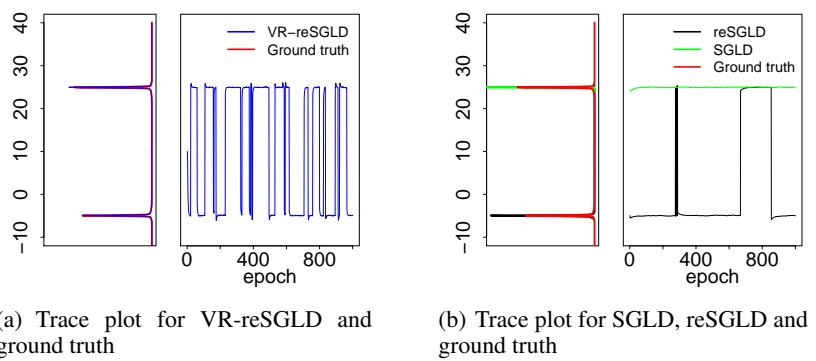

(a) Trace plot for VR-reSGLD and ground truth

(b) Trace plot for SGLD, reSGLD and ground truth

Figure 6: Trace plots and KDEs of $\beta^{(1)}$

with long enough iterations. In Figs 6(a) and 6(b), we observe that, even though the distribution of interest has a pickier density, our proposed algorithm VR-reSGLD was able to detect both modes and acceptably jump between them. On the other hand, the competitor algorithm SGLD was trapped in the first mode visited and never escaped. reSGLD was able to jump some times between modes only after considering a substantial factor $F = 500$ which, according to the theory, introduces bias.

