# OpenReview forum: "Accelerating Convergence of Replica Exchange Stochastic Gradient MCMC via Variance Reduction"
_ICLR.cc/2021/Conference — ICLR 2021 Poster_

### Official Review · AnonReviewer4 · 2020-10-26
**Paper could be accepted after a thorough review.**

**Rating:** 6
**Confidence:** 3

**Review:**

# Summary

The paper presents VR-reSGLD, a method to accelerate replica exchange stochastic gradient Langevin diffusion (reSGLD), which has been proposed recently to tackle non-convex learning problems. reSGLD suffers from two major sources of error resulting in low swapping rates: minibatch noise and the discretization error of Langevin diffusion. The idea of the paper is to use control variates to reduce the variance of energy estimators and thereby improve the swapping rate (which should lead to an accelerated convergence). Unlike previous modifications of SGLD, the variance reduction proposed in the paper aims at improving the energy estimators rather than the gradient estimators. The paper presents non-asymptotic results backing the intended acceleration of the Markov jump process. Numerical experiments illustrate the performance gain achieved by VR-reSGLD. These tests include a one-dimensional example (learning the component mean of a bimodal mixture of Gaussians) and Bayesian training of DNNs based on CIFAR imaging data.

# Assessment

Overall, I think the paper could be accepted in principle, but requires a thorough revision.

reSGLD has been proposed quite recently (ICML 2020); and the use of variance reduction (VR) techniques is common to account for minibatch noise. So the combination of reSGLD and VR seems like an obvious idea. However, the current paper adds more than just a minor modification of reSGLD in that it provides interesting theoretical results and illustrates the validity of these results by numerical experiments.

## Pros

1. Convergence in 2-Wasserstein distance, no asymptotic normality assumed.

2. Significant acceleration, tighter discretization error.

3. Improved uncertainty quantification.

## Cons

1. The theoretical part of the paper is hard to follow for a non-expert.

2. There are quite a number of algorithmic parameters (e.g. $m$, $n$, $\eta$, $\gamma$, $F$, etc.) that seem to require some tweaking.

3. The improvement presented on imaging data (Table 1) seems to be rather marginal.

# Comments and Questions:

- How much tweaking do parameters such as $F$ etc. require? Can you provide some intuition on how to set these parameters?

- Have you considered learning rates $\eta_k$ that depend on the temperature?

- Have you considered using more than two temperatures?

- Please define "acceleration" (relative to what?)

- Page 2: "To avoid the crude estimate, ..." is unclear. What is "the crude estimate"? The upper bound provided by Gronwall's inequality?

- Page 3: The sentence starting with "The underlying Dirichlet form ..." is a bit obscure. Please try to clarify.

- Page 3: Regarding the constant $c$ achieving minimum variance: Is the sign of $c^\star$ correct? It seems that the denominator should be $\text{Var}(B|\hat\beta^{(h)})$ rather than $\text{Var}(B|\beta^{(h)})$.

- Page 4, Algorithm 1: How to set $\gamma$? The update rule for $\tilde \sigma_k^2$ is unclear to me. What is $\sigma_k^2$? Should this be $\sigma^2$?

- Page 5, Theorem 1: What do you mean by "exponentially faster", compared to reSGLD? What are the $\mathcal{E}$s?

- Page 6: Your comment after Theorem 3 ("This theorem implies ..."). Why does Theorem 3 imply a much larger swapping rate?

- Page 6: You set $\tau^{(1)}=10$ "to avoid peaky modes", but it would be interesting to see the performance of VR-reSGLD for a sharply peaked posterior (which is more realistic when thinking about high-dimensional parameter spaces and a large number of data...)

- Page 7: Why do you anneal $\tau^{(1)}$ in the CIFAR experiment?

# Minor:

## Typos

- Replace "Gröwall" with "Grönwall" or "Gronwall"

## Grammar and wording

* In "obtain the state-of-the-art results" delete "the"

* Try to improve: "the noisy energy estimators in mini-batch settings render the naive implementation a large bias..."

* Page 2: "it may cause the process to stuck in ..."

* There a probably more problematic phrases. Please improve the quality of writing.

---

> ### Author Response · Authors · 2020-11-17
> **Hyper-parameters, multiple temperatures, and others**
>
> We appreciate the detailed and valuable comments.
>
> Q1: How much tweaking do these parameters such as $F, m, n, \eta, \gamma, \tau$ require?
>
> $F$ is an important hyperparameter and the tuning directly affects the empirical swapping rates. We would like to study the extension of a more user-friendly replica exchange algorithm directly based on the target swapping rates in the future.
>
> The other hyperparameters don't require much tuning. The hyperparameter settings of $n, \eta$ and $\tau^{(1)}$ can be naturally imported from SGD and SGHMC. $\gamma$ is a smoothing factor and can be fixed at $0.3$ by default. The update frequency $m$ can be chosen in the order of batch size $n$ and we can try $\tau^{(2)}$ in $\{3\tau^{(1)}, 10\tau^{(1)}, 30\tau^{(1)}, 100 \tau^{(1)}\}$.
>
> Q2: Consider more than two temperatures?
>
> Yes, it is natural to study the population version with many temperatures. However, the naive extension may lead to little communication between the chains at the two ends of temperatures, which hence deteriorates the performance. To solve this issue and improve the efficiency of parallel tempering, we may consider optimizing the temperature intervals [1], adapting the number of sweeps to the canonical autocorrelation time [2], and exploiting the even-odd swapping scheme to optimize the round trip rate [3].
>
> We leave the systematic study of population parallel tempering with the minimal possible hyperparameters in the future.
>
>
> [1] Katzgraber, etc. Feedback-optimized Parallel Tempering Monte Carlo. J. Stat. Mech. 2006
>
> [2] Bittner, etc. Make Life Simple: Unleash the Full Power of the Parallel Tempering Algorithm. Phy. Rev. Let. 2008.
>
> [3] Syed, etc.  Non-Reversible Parallel Tempering: A Scalable Highly Parallel MCMC scheme. arXiv:1905.02939v2, 2019.
>
> Q3: Define acceleration (relative to what?) and what do you mean by "exponentially faster" on page 5, compared to reSGLD?
>
> The acceleration means to accelerate the exponential convergence of the continuous-time process of the standard replica exchange SGLD. We have rephrased the words in the revised paper to make the statement clear and rigorous.
>
> Q4: What is "the crude estimate"? The upper bound provided by Gronwall's inequality?
>
> Yes.
>
> Q5: "The underlying Dirichlet form ..." is a bit obscure and what is the $\cal E$
>
> We have included discussions of the Dirichlet form in the preliminary section and also linked it to Theorem 1.
>
> Q6: Page 3: The sign of $c^{\star}$ is not correct and the denominator should be $\mathrm{Var}(B|\widehat \beta^{(h)})$. Page 4: The update rule for $\tilde \sigma_k^2$ is unclear.
>
> Thanks for pointing out the typos. We have corrected them in the revised paper. The code implementation was not affected.
>
> Q7: Page 4, Algorithm 1: How to set $\gamma$?
>
> $\gamma$ is a smoothing factor to filter out high-frequency noise. A larger $\gamma$ captures the trend better but is less robust. Empirically, we can set $\gamma=0.3$ by default.
>
> Q8: Page 6: Why does Theorem 3 imply a much larger swapping rate?
>
> We acknowledge that a much larger swapping rate should better be stated after Lemma 2. We have rephrased the comments in the revision.
>
> Q9: Page 6: You set $\tau^{(1)}=10$ to avoid peaky modes, but it would be interesting to see the performance of VR-reSGLD for a sharply peaked posterior (high-dimensional parameter spaces and a large number of data).
>
> Extension of $\tau^{(1)}$ to a different value such as 1 is straightforward. As shown in section G in the appendix of the revised paper, a similar conclusion still holds. However, the posterior may not be always peaked in high-dimensional big data problems. For example, as indicated by some discussions [1,2,3], sharply peaked optima may generalize worse than wide optima.
>
>
> [1] Keskar, etc. On Large-Batch Training for Deep Learning: Generalization Gap and Sharp Minima. ICLR'17.
>
> [2] Chaudhari, etc. Entropy-SGD: Biasing Gradient Descent into Wide Valleys. ICLR'17.
>
> [3] Li, etc. Visualizing the Loss Landscape of Neural Nets. NeurIPS'18.
>
> Q10: Page 7: Why do you anneal $\tau^{(1)}$ in the CIFAR experiment?
>
> We annealed $\tau^{(1)}$ because it could provide roughly a 0.3\% improvement on the predictions of CIFAR100. In the code we attached in the supplementary file, we tried a slightly new strategy and only annealed the temperature during the warm-up period, the performance was almost the same as before and no annealing was conducted during the Bayesian-model-averaging period. We will give a comprehensive study of this strategy in the next revision.

---

### Official Review · AnonReviewer3 · 2020-10-27
**Nice variant of reSGLD using Variance Reduction on energy estimators**

**Rating:** 7
**Confidence:** 3

**Review:**

The authors propose a variant of the Replica Exchange Stochastic Gradient Langevin Dynamics (reSGLD) for non log-concave sampling by using a variant reduction technique on the estimation of the swapping rate. Assuming that the log-density is a finite sum. the authors apply classical variance reduction techniques to the energy estimator necessay to compute the swapping rate. They show that applying such technique yields a higher swapping frequency and faster convergent rate of both the continuous time SDE and its dicretization scheme. Finally, the authors perform numerical experiments on both synthetic and real world data, and show that VR indeed reduces the variance of energy estimator by several orders of magnitude, hence inducing faster convergence.

In the algorithm, what is the role of the thining factor T? Is it involved in the convergence guarantee given by Theorem 3?

For the synthetic experiments, can you explain this choice for different values of F (1 for VR-reSGLD and 100 for reSGLD)? I can believe that variance reduction lkeads to a different bias/variance trade-off, but it would be good to explain whythese values of F were used.

Thm 1: Operator E is not defined in the main text (only in the appendix).

---

> ### Author Response · Authors · 2020-11-17
> **The role of the thining factor T and why different values of F are used**
>
> We appreciate the valuable comments.
>
> Q1: What is the role of the thinning factor T? Is it involved in the convergence given by Theorem 3?
>
> The main purpose of a thinning factor is to avoid a cumbersome system and extensive computations in Bayesian model averaging and the reduced computations yield an improved statistical efficiency [1]. It is not directly related to the convergence guarantee given by Theorem 3.
>
> [1] Owen. Statistically efficient thinning of a Markov chain sampler. Technical Report. 2015.
>
> Q2: For the synthetic experiments, why different values of F are used (1 for VR-reSGLD and 100 for reSGLD)?
>
> It is ideal to set F=1 if there are enough swaps. However, such a choice leads to no swaps for the standard reSGLD in the synthetic experiment, hence a larger $F$ is applied to reSGLD and only leads to several swaps. By contrast, VR-reSLD doesn't have this issue due to the significant reduction of variance and setting F=1 still yields sufficiently many swaps.
>
> Q3: Thm 1: E is not defined in the main text.
>
> We have included discussions of the Dirichlet form in the preliminary section and also linked it to Theorem 1.

---

### Official Review · AnonReviewer1 · 2020-10-28
**Interesting, well written, limited originality**

**Rating:** 7
**Confidence:** 3

**Review:**

Accelerating convergence of replica exchange stochastic gradient mcmc via variance reduction

Summary:

The paper presents a variance reduction technique to achieve more efficient swaps in replica exchange stochastic gradient Langevin dynamics MCMC. The paper provides detailed analysis of the method as well as empirical evaluation on some standard deep learning tasks.

Positive:

1. Overall I would say that the paper is well written and and it is fairly easy to follow the presentation and details in the derivations.
2. The topic is very timely and the method appears to be very useful. As an attractive method for minibatched Bayesian inference, stochastic gradient Langevin Dynamics samplers are of high interest, but tuning the algorithm can be somewhat finicky in my experience. Replica exchange is sometimes extremely useful, and finding good defaults for these types of methods is important.
3. Experimental validation is reasonable (although a bit limited) and the methods chosen for comparison are resonable.
4. A comprehensive set of appendices are included to provide further details. Although I did not go through the appendices in detail, I find it appealing that further information is provided for readers wishing to apply these methods in practice.

Negative:

1. The authors do not provide a reference software implementation. This makes it more difficult for readers to verify the results and might limit the impact of the paper. I would highly appreciate that the authors would create and share a minimal implementation.
2. The novelty / originality is limited: A well known type of variance reduction applied in a new way/context where it makes perfect sense though.

Recommendation:

Good paper. Accept.

---

> ### Author Response · Authors · 2020-11-17
> **Software implementation and novelty**
>
> We appreciate the valuable comments.
>
> Q1: I would highly appreciate that the authors would create and share a minimal implementation.
>
> We have attached the code for section 5.2 in the supplementary file. We will release all of them in the final version.
>
> Q2: Limited novelty in variance reduction.
>
> We acknowledge the main variance reduction method is quite standard, which, however, suffers from an insufficient reduction of variance when the learning rate is large and the correlation is weak. To handle this issue, we further proposed the adaptive variance-reduced replica exchange SGLD in Algorithm 2 in the appendix to adaptively estimate the unknown optimal correlation. As a result, we obtained around 40% improvement in variance reduction in the early stage.
>
> We leave the study of more powerful variance reduction techniques, such as multiple reference points/ control variates [1], lightweight SAGA in the future.
>
> [1] Dongruo Zhou, etc. Stochastic Nested Variance Reduction for Nonconvex Optimization. JMLR'19.

---

### Official Review · AnonReviewer2 · 2020-10-29
**Proposed the control variates based variance reduction technique for replica SG-MCMC. The analysis seems new.**

**Rating:** 5
**Confidence:** 3

**Review:**

##  Summary of the paper
This paper extends the replica stochastic gradient MCMC by incorporating the new variance reduction technique. The author analyzed the non-asymptotic theoretical behavior of the proposed method, which shows the proposed method provides a better energy landscape.

## Strong and weak points of the paper
### Strong points
- Although the applied variance reduction technique is not new, the analysis of the swapping rate in Lemma 2 seems novel compared to the past replica-exchange MCMC work.
- Provided an asymptotic analysis for the variance reduction in Theorem 4.

### Weak points
- Proposed variance reduction is not new, I think. It is very similar to the standard control variates methods, which had been extensively applied into MCMCs and other machine learning tasks.

## Rating
- Clarity: Well written, easy to read, although I did not check the proofs in detail.
- Correctness: I did not check all the proof in detail.
- Novelty: The idea seems not novel but the swapping rate analysis seems new and interesting.

## Comments and Questions
- Q) All the experimental results and theoretical analysis suggest that using a smaller $m$ is better. But how can we tune this $m$? Does this variance reduction work even I use $m=1$?  But I also think that sufficiently large $m$ is required to estimate the control covariate appropriately as the work of  Rie Johnson and Tong Zhang suggest in their paper. Does this intuition wrong?

- Q) In Sec5.2, as for Fig.3 c and d, the effect of variance reduction becomes significant after the sufficient epochs and the author conjectured that this is because that the learning rate is decreased. I think that this suggests that the proposed variance reduction is significantly affected by the step size which changes during the exploration stage in SG-MCMCs. And thus I thought that using the constant $m$ during the exploration stage might not be a good idea.

---

> ### Author Response · Authors · 2020-11-17
> **Setting m in the order of  n gives a good empirical performance**
>
> We appreciate the valuable comments.
>
> Q1: How to tune $m$?
>
> The key to tuning $m$ is to balance between acceleration and cost. For example, choosing $m$ in the order of $n$ gives a good empirical performance on CIFAR10 and CIFAR100 datasets. Further reducing $m$ may yield a diminishing marginal utility with a high cost in gradient evaluation.
>
> Q2: Does $m=1$ work in variance reduction?
>
> Yes. In such a case, variance reduction works perfectly because the exact energy is used in each iteration. Nevertheless, the computational cost may be unacceptable.
>
> Q3: Why Johnson and Zhang [1] suggest a sufficiently large $m$?
>
> We focus on different theoretical aspects. [1] aims to prove a geometric convergence in convex optimization for a large value of $m$; our work concentrates on the variance reduction of energy estimators to propose more effective swaps to accelerate the exponential convergence of the continuous-time dynamics in non-convex learning. Similar theoretical results that a smaller $m$ yields a smaller variance have been obtained in Theorem 3 [2].
>
> To be more specific, Theorem 1 of [1] only works in a manner of every $m$ iterations. The second inequality in page 5 [1] is essentially $\mathop{\mathbb{E}}[P(w_{k+m})-P(w_{\star})]\leq \alpha \mathop{\mathbb{E}}[\left(P(w_{k})-P(w_{\star})\right)]$. In the extreme case $m=\infty$, the theorem only guarantees that the output after an infinite number of iterations is better than the initial point.
>
> [1] Johnson and Zhang. Accelerating Stochastic Gradient Descent using Predictive Variance reduction. NIPS'13.
>
> [2] Dubey, etc. Variance Reduction in Stochastic Gradient Langevin Dynamics. NIPS'16.
>
> Q4: In Sec5.2, the author conjectured that the effect of variance reduction becomes significant because the learning rate is decreased. I think the proposed variance reduction is significantly affected by the step size and using the constant might not be a good idea.
>
> Our experimental result agrees with the theoretical analysis in Lemma 1, which says that a smaller learning rate leads to a better variance reduction given the same update frequency $m$ and batch size $n$. Using a constant $m$ is quite standard in variance reduction, although we acknowledge the potential benefit of an adaptive $m$ in improving the performance of the proposed algorithm. We would like to study the extension of an adaptive $m$ in the future.

---

### Decision · Program_Chairs · 2021-01-07
**Final Decision**

**Decision:**

Accept (Poster)

**Comment:**

This work aims at doing Bayesian inference via Langevin dynamics with data subsampling. This builds on previous work with "replica exchange" where parallel chains are run at different temperatures and can be swapped to encourage moving between modes. The main technical novelty here is a scheme to reduce variance. This is done in the style of SGRD by periodically computing the gradient on all data and then using those values as control variates. This is shown to reduce variance.

Reviewers generally felt that this represented a sensible combination of known ideas aimed at an important and timely problem with sufficient empirical evaluation. There was consensus the paper was clearly written. I concur that even if the combination is "expected" to work, the presence of guarantees for performance represent sufficient technical novelty. I particularly applaud the fact that the paper does not over-claim and generously gives credit to related work. This is helpful to the reader and encourages the flow of ideas. For these reasons I recommend acceptance of the paper.

In reading the paper, I had a couple questions about the experiments:

1. It's not obvious to me from the experiments how specific the method is to the replica exchange setting. The main control variate idea appears to be applicable without replica exchange. I would very much like to see a "VR-SGHMC" row in Table 1 unless there is a good reason that this cannot be done. It would be very beneficial to understand the contributions of these different algorithmic components.

2. The CIFAR experiments directly test variance. That's fine, the paper is aimed at reducing variance, after all. However, I would like to see more tests of the follow-on improvements in optimization speed. It has been my experience that improvements in variance sometimes produce surprisingly small improvements in optimization speed. My intuition for this is that reduced variance mostly helps by making it possible to use a larger step-size without the same penalty in the stationary dist. In practice, the step-size typically ends up being imperfect, meaning that changes in variance have small changes.

---

> ### Comment · ~Wei_Deng2 · 2021-03-05
> **Energy variance reduction accelerates the exponential convergence**
>
> We appreciate your suggestions.
>
> We tried to train ResNet20 on CIFAR100 based on VR-SGHMC with variance-reduced gradients and the default learning rate and found that it performs even worse than the vanilla SGHMC. We suspect that the learning rate is well-tuned for SGHMC but sub-optimal for VR-SGHMC. To improve the performance, VR-SGHMC requires the additional tuning of learning rates, while ours does not.
>
> As to the optimization speed, we tried to run reSGHMC and cycSGHMC on ResNet32 with the same time budget as VR-reSGHMC and obtained around 76.9\%$\pm$0.3\% and 77\%$\pm$0.2\% accuracy, respectively, which is still not better than 77.4\%$\pm$0.3\% based on our method. This shows the potential of energy variance reduction in improving the optimization speed. We conjecture that the reason why energy variance reduction works well is it not only reduces the discretization error in the swaps (and accepts larger swapping intensity) but also accelerates the exponential convergence.